# Large-scale risk assessment on snow avalanche hazard in alpine regions

Gregor Ortner[1,2,3], Michael Bründl[1,2], Chahan M. Kropf[3,4], Thomas Röösli[3,4], Yves Bühler[1,2], and David N. Bresch[3,4]

[1]WSL Institute for Snow and Avalanche Research SLF, 7260 Davos Dorf, Switzerland
[2]Climate Change, Extremes and Natural Hazards in Alpine Regions Research Center CERC, 7260 Davos Dorf, Switzerland
[3]Institute for Environmental Decisions, ETH Zurich, Universitätstr. 16, 8092 Zurich, Switzerland
[4]Federal Office of Meteorology and Climatology MeteoSwiss, Operation Center 1, P.O. Box 257, 8058 Zurich-Airport, Switzerland

**Correspondence:** Gregor Ortner (gregor.ortner@slf.ch)

**Abstract.** Snow avalanches are recurring natural hazards that affect the population and transport infrastructure in mountainous regions during the winter months such as in the most recent avalanche winters of 2018 and 2019, where large damages were caused by avalanches throughout the Alps. Decision makers need detailed information on the spatial distribution of the hazard and risk to prioritize and apply appropriate adaptation strategies and mitigation measures to minimize impacts. Here, we present a novel risk assessment approach for assessing the spatial distribution of avalanche risk by combining large-scale hazard mapping with a state-of-the-art risk assessment tool, where risk is understood as the product of hazard, exposure, and vulnerability. Hazard disposition is modeled using the large-scale hazard indication mapping method RAMMS::LSHIM, and risks are assessed using the probabilistic Python-based risk assessment platform CLIMADA, developed at ETH Zürich. The avalanche hazard mapping for scenarios with a 30-, 100-, and 300-year return period is based on a high-resolution terrain model, 3-day snow depth increase, automatically determined potential release areas, and protection forest information. Avalanche hazard for 40,000 single snow avalanches is expressed as avalanche intensity measured as pressure. Exposure is represented with a detailed building layer indicating the spatial distribution of monetary assets. Vulnerability of the buildings is defined by damage functions based on the software EconoMe, which is in operational use in Switzerland. The outputs of the hazard, exposure, and vulnerability analyses are combined to quantify the risk in spatially explicit risk maps. The risk considers the probability and intensity of snow avalanche occurrence as well as the concentration of vulnerable, exposed buildings. Uncertainty and sensitivity analyses were performed to capture inherent variability in the input parameters. This new risk assessment approach allows for the quantification of avalanche risk on large scales and results in maps that show the spatial distribution of risk at specific locations. Large-scale risk maps can assist decision makers in identifying areas where hazard mitigation and/or adaption is needed to address current and future avalanche risk.

# 1  Introduction

In a time of densely populated mountainous landscapes and continuous socio-economic growth, society is increasingly exposed to natural hazards (Zgheib et al., 2020). In mountainous regions such as the Alps, avalanches are a significant natural hazard in winter causing damage to buildings and infrastructure. In the past 20 years, countries situated in the European Alps such as Switzerland have experienced multiple catastrophic avalanche situations. The winter 2017/18 was the first since the catastrophic avalanche winter of 1999 (Wiesinger and Adams, 2007) in which the highest avalanche danger, level 5, was forecasted for wide areas across the Swiss Alps. In January 2018, 2.5 to 5m of snow fell at higher altitudes within 25 days. Numerous avalanches in the categories "large" and "very large" were counted (Zweifel et al., 2019). In total, over 380 avalanches caused a damage to buildings, traffic routes, or important infrastructure in Switzerland (Bründl et al., 2019), making it to the most severe avalanche winter in recent years, not only in Switzerland, but also in Austria, and Germany (MeteoSchweiz, 2019; Pancevski, 2019; ZAMG, 2020; Trachsel et al., 2020). Also in winter 2018/19, for example, exceptional snowfall events occurred which caused high damages throughout Switzerland (Trachsel et al., 2020). Such events show that even in highly developed, well adapted countries, society is still vulnerable to extreme snowfall events causing avalanche hazards. Strategies, methods, and risk assessments to counteract this threat are well developed in most areas in the Alps, but they need to be continuously developed to strengthen and improve the resilience of the population and their assets (Zgheib et al., 2020).

To cope with natural hazards threatening exposed assets, various institutions have introduced the concept of risk to the field of natural hazards. The IPCC for instance defines risk as the likelihood for the disturbance of the normal functionality of a society due to a hazardous physical event under vulnerable social conditions (= vulnerability) with economic, material or environmental consequences (IPCC, 2014), see also Fig. 1. According to this definition, the risk concept can also be applied to harmful consequences to the environment and environmental systems. A consideration of the impacts of natural hazards to the environment is generally also of importance to the authors of this study. But since the damage to human life's and man-made objects is of greater importance to society and decision makers, this concept is mostly applied to infrastructure and buildings, as it is also in this study.

The concept of risk has been introduced in Switzerland in the late 1990s to support decision makers for dealing with natural hazards (Heinimann et al., 1998; Borter, 1999; Bründl et al., 2009). The risk concept became also the central element of the Strategy for Natural Hazards of the National Platform Natural Hazards (PLANAT) in Switzerland (PLANAT, 2009, 2018). Overviews on natural hazards risks and vulnerabilities in different spatial scales were generated by various institutions in the last decade (MATRIX Consortium, 2013; van Westen and Greiving, 2017; Fuchs et al., 2019; FOEN, 2020). These studies vary from vulnerability surveys (Fuchs et al., 2015) to multi-risk and resilience approaches (Kappes et al., 2012; Komendantova et al., 2016). Projects such as RoadRisk carried out by the Swiss Federal Roads Office (ASTRA, 2012) or the National Risk Overview initiated by the Swiss Federal Office for the Environment (FOEN, 2020) were created with a lot of effort and long project duration's. Such projects indicate the demand for large-scale risk surveys.

What is missing so far is a method to assess avalanche risk at a regional or national scale, with a state-of-the-art hazard mapping tool, which would allow decision-makers to identify hot spots of avalanche risk. The so-produced risk maps show in short term

studies where detailed assessment would be necessary to develop appropriate risk management.

The goal of this study is to suggest a framework for assessing avalanche risk at a large scale, as it is presented in Fig. 1. The method was applied in a case study to a region in central Switzerland, but could be deployed anywhere in the world. Since the components of risk are not constant, such a framework can also help to depict changes of the components of risk, such as hazard, exposure and vulnerability over time and space. In this paper, we focus on the presentation of the framework for the current risk situation, which will serve as a platform for modelling expected climate and socio-economic induced changes of

risk in future.

Especially in the context of climate change and population growth, new strategies and tools that assist to systematically identify risk and respond to threats in exposed areas are of increasing importance (IPCC, 2012). Changes in the climate system and their influence on local weather phenomena do not only affect us already (CH2018, 2018), but will likely lead to an increase of the frequency and magnitude of natural hazards in the years to come (IPCC, 2014). In particular, various studies indicate that

changes in the climate system, such as temperature rise and an increase of extreme precipitation events, will likely influence gravity-driven hazards (Mani and Caduff, 2012; Ballesteros-Cánovas et al., 2018), such as snow avalanches.

The paper is organised as follows. First, we present the methods in detail followed by a section in which we show how these new methods operate and how they can be applied to specific examples, such as the case study in central Switzerland. In the results section, we depict the spatial distribution of the calculated risks, analyse the uncertainties and perform a sensitivity

study. Finally, we discuss progress and limitations of the new method and conclude, how this framework might contribute to a dialogue on the changes of risk at a large scale.

## 2 Methods

### 2.1 Risk

In the context of natural hazards in Switzerland, risk is defined as the product of hazard potential, objects at risk (exposure) and

their vulnerability (Borter and Bart, 1999). In more detail, Bründl et al. (2016), defines risk as the damage that is statistically expected due to the hazard intensity (= caused by avalanche pressure) in a given scenario, calculated as the product of the expected damage and the frequency (= 1/return period) of this scenario.

In this study, the risk tool CLIMADA is used for risk assessment and therefore we use the definition of the framework developers. It is a similar but extended definition of the IPCC risk concept by Aznar-Siguan and Bresch (2019) expressing risk as

the probability of a consequence resulting from a hazard and its severity:

$$\text{risk} = \text{probability} \times \text{severity} \tag{1}$$

, where

$$\text{severity} = F(\text{hazard intensity, exposure, vulnerability}) = \text{exposure} \times \text{vulnerability function(hazard intensity)} \tag{2}$$

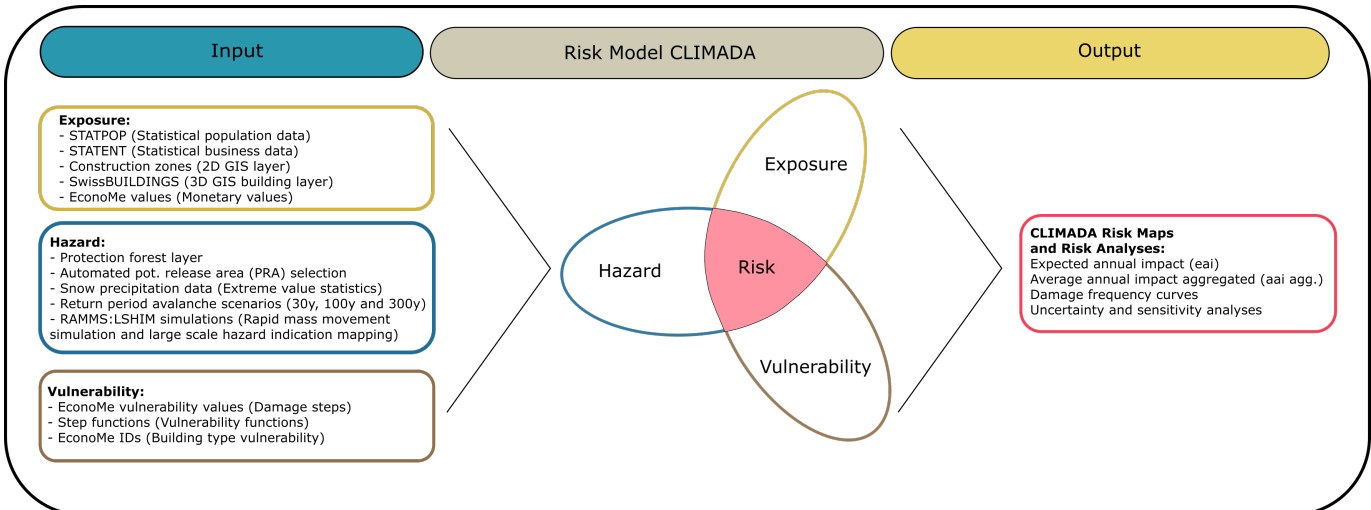

**Figure 1.** A schematic illustration of the applied IPCC risk concept with the used input components for generating the base information on exposure, hazard and vulnerability, used to create risk maps and perform risk analyses.

Since the consideration of risk requires the three components of **Hazard**, **Exposure** and **Vulnerability**, see Fig. 1, we give a brief introduction to this components: The **Hazard** (Section 2.3) class is obtained by avalanche hazard mapping using the "RAMMS:LSIHM" (Rapid Alpine Mass Movement Simulation:: Large Scale Hazard Indication Mapping) method. For this purpose, a protection forest layer (Section 2.3.2) is generated that defines where protection forest is located. Using this layer, potential release areas (PRA) can be identified with an automatic algorithm. Using extreme value statistics, maximum 3-day snowfalls are analysed and three avalanche scenarios with return periods (rp) of 30 years, 100 years and 300 years are defined (Section 2.3.1). PRA calculation and also protection forest creation take these scenarios into account. With the defined potential release areas and an assigned amount of snow, the RAMMS (Section 2.3.4) avalanche simulations can now be carried out and a hazard indication map can be generated as a hazard class.

The **Exposure** (Section 2.4) is a data set that defines the exposed monetary values. It is composed of statistical data on the resident population called STATPOP (number of persons per building) and the economic data STATENT (location of places of business). These are available in a GIS data set with coordinates. Furthermore, Swiss building zones and a 3D building data set are used. Monetary values are then assigned using the Swiss standard risk methodology "EconoMe" (Section 2.4.1). Thus, a new exposure data set is created that shows the geographical distribution of the monetary values on a map (Section 2.4.2).

The third component is the **Vulnerability** (Section 2.5) class, which defines the damage sensitivity of an object that is exposed to a certain avalanche pressure. The EconoMe risk methodology specifies values for building damage in percentage caused by a certain avalanche pressure. These values are adopted in steps and define the step function as a so-called impact or vulnerability

function (Section 2.5.1). According to EconoMe, these functions can be assigned to a certain building type by means of an ID and thus be used in the risk framework.

To express these components in numbers in the context of the **Risk** concept (Section 2.1), the CLIMADA (Section 2.1.1) (Climate Adaptation) risk model is used. This model allows us to express the risk in monetary terms such as the expected annual impact (eai) as well as the aggregated average annual impact (aai agg.) and depict them on spatial risk maps (Section 3.1) of the considered region. These values can also be presented in damage frequency curves (Section 3.3) to show at which return period which monetary losses would be expected. A sensitivity and uncertainty analysis (Section 2.6.1) completes the

risk assessment and shows the limitations of the framework.

The methods chapter is organized into subsections explaining all components of the risk framework from hazard, exposure and vulnerability to risk, and its application to the case study region (Section 2.2) in detail. These subsections contain the information on how the respective components were defined and generated in this study and how they are used to calculate the spatially distributed monetary terms of risk.

### 2.1.1 CLIMADA

CLIMADA is an open-source and -access *Python* package for probabilistic risk assessment (Aznar-Siguan and Bresch, 2019). It allows for the computation of the impact of natural hazards, modelled as intensity maps, on exposures, modelled as value maps, considering the vulnerability of the exposures, modelled as impact functions. It is possible to compute the risk today and in the future, including climate change that modifies the hazard and socio-economic development affecting the exposures and

vulnerability. Finally, one can compute the reduction in risk and cost from adaptation options to perform a cost-benefit analysis (Kropf et al., 2021). For all the model outputs, CLIMADA provides a module to perform uncertainty and sensitivity analysis (Kropf et al., 2022) using global (quasi-)Monte-Carlo sampling.

In this project, we use CLIMADA to compute the risk of avalanches to more than 13,000 individual single buildings, from each of the 40,000 simulated avalanches using the hazard intensity maps described in Sect. 2.3, the exposures distribution

described in Sec. 2.4 and the impact functions introduced in Sec. 2.5. As defined in the CLIMADA tool (Aznar-Siguan and Bresch, 2019), the impact is expressed by the following risk quantities:

- the expected annual impact (eai) for each exposure (affected building) expressed in monetary terms per year (CHF per year). It combines the hazard intensity with its expectation (= return period) and the damage degree from the impact functions;

- the average annual impact (aai) is the average of the expected annual impacts (eai) over all exposures, expressed in monetary terms per year (CHF per year);

- the average annual impact aggregated (aai agg.) is the sum of the average annual impacts (aai) for each scenario or all scenarios combined expressed in monetary terms per year (CHF per year). The aai agg. corresponds to the overall risk.

To show general patterns of risk and to look at the overall risk of all scenarios, we adjusted the three scenarios for the

combination with regard to the return period by $p_{30} = p_{30} - p_{100} = 1/30 - 1/100 = 0.0233$ and $p_{100} = p_{100} - p_{300} = 1/100 -$

$1/300 = 0.00667$ and $p_{300} = 1/300 = 0.0033$. This allowed us to combine the hazard scenarios and recalculate the overall impacts such as the expected annual impact (= eai) for single objects over all hazard scenarios in the entire study region as it is shown in the result section.

In a further step, the uncertainties and sensitivity of the risk framework will be derived as described in Sec. 2.6.

## 2.2 Case study region

For this study we focus on a case study area of the Gotthard region (Fig. 2) in central Switzerland between Altdorf and Göschenen in the Swiss Canton of Uri, one of the most important north-south transit corridors in Europe. The Gotthard area is one of the snowiest regions in Switzerland and characterized by many steep, non-forested slopes of different orientations making the region highly exposed to avalanches. Steep meadows or rocky slopes form avalanche release areas from 28° to 50° slopes in the upper catchment area of the main Reuss valley and reach from an altitude of approx. 1700 m a.s.l. up to almost 3000m a.s.l. At heavy snowfall events, avalanches start in the backcountry release zones and sometimes flow down to valley bottom where the snow masses are deposited in the river bed of the Reuss river.

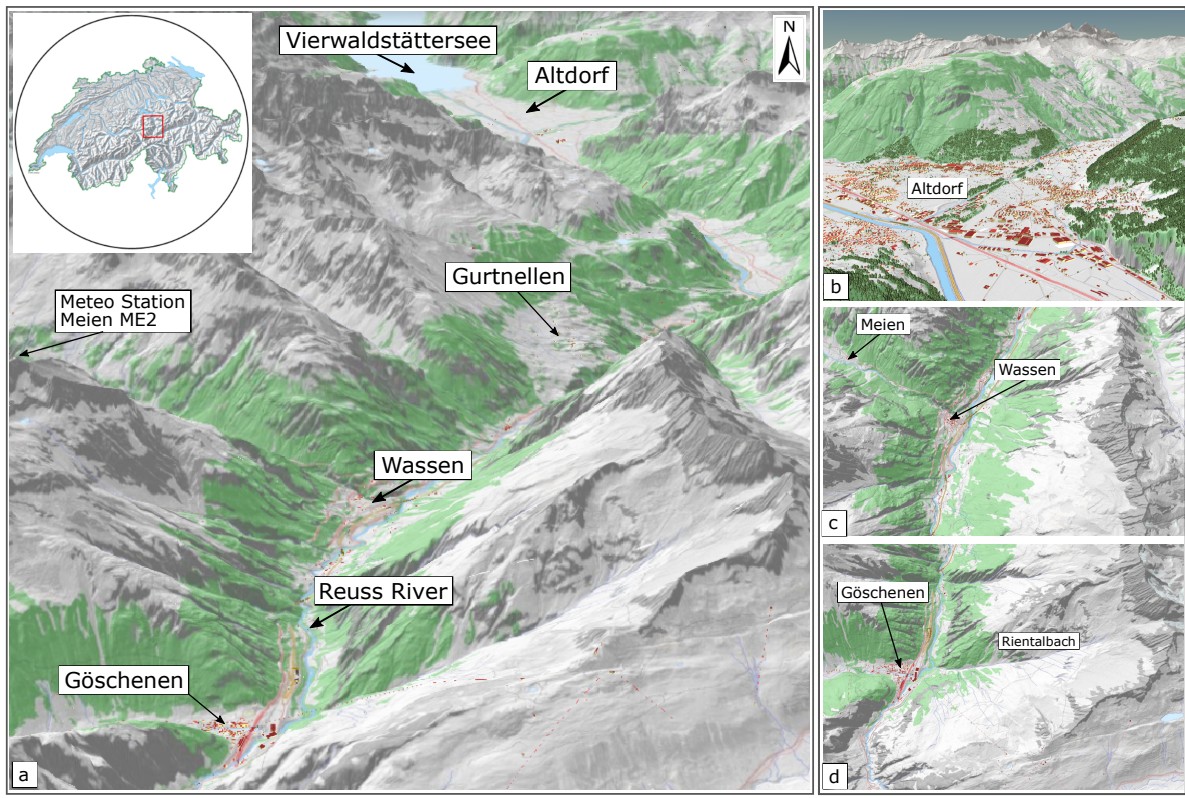

**Figure 2.** Illustration of the case study area of the Gotthard region in central Switzerland with the avalanche prone slopes. a) overview of the region with a map panel of Switzerland. b) Detail of the village Göschenen with the well known Rientalbach avalanche path. c) Hazard hot-spot Wassen. d) Altdorf and its surrounding slopes. Forests are shown in green and buildings in red. Map base data source: Swiss Federal Office of Topography (map.geo.admin.ch)

## 2.3 Hazard - Avalanche hazard indication mapping with RAMMS::LSHIM

Hazard indication mapping of snow avalanches at a scale of 1:5,000 to 1:10,000 was carried out for backcountry users (Harvey et al., 2018) and exposed communities in Switzerland (Bühler et al., 2022), Italy (Maggioni et al., 2018; Monti et al., 2018), and other regions worldwide (Bühler et al., 2018). To cover all possible avalanches affecting the region, every hydrological catchment stretching from the main valley up to the mountain ridges of the side valleys was regarded as potential process area. The individual catchments were combined into a large comprehensive perimeter outlining the study area (Fig. 4). To identify relevant catchment areas that are particularly affected by avalanches, a data set of all historical damaging avalanches observed and recorded by the WSL Institute for Snow and Avalanche Research SLF and cantonal and federal authorities were taken as a basis. The selection of the hydrological catchment areas also includes areas with no avalanche records but were regarded as relevant for hazard indication mapping in this study.

### 2.3.1 Definition of three avalanche scenarios

To cover a large range of potential avalanche events, we consider 3 different avalanche scenarios (see Tab. 1). One frequent scenario [A] corresponding to a snowfall event with a 30-year return period, one intermediate [B] (100-year return period) and one extreme [C] scenario (300-year return period). The definition of the scenarios is implemented according to the maximum 3-day snow depth increase and directly determines the mean fracture depths, thereby the release volume and thus the modelled avalanche run out lengths. The higher the return period, the larger the mean avalanche volume. The term return period (rp) describes the average number of years between two comparable events of the same intensity at the same location. By hazard frequency we denote the probability of occurrence calculated as reciprocal value of the return period (1/rp). This definition is common practice in avalanche hazard mapping (PLANAT, 2009; Salm et al., 1990).

The derivation of the 3-day snow depth increase was based on long-term snow measurement series at meteorological stations representative for the chosen area. As shown in Fig. 3, two extreme value statistical methods were applied: the generalized extreme value distribution with maximum likelihood estimations (GEV - MLE) and the Gumbel distribution (Bocchiola et al., 2008) with maximum likelihood estimations (GUM-MLE). Table 1 shows that both methods produce similar values, but slightly higher values with Gumbel (GUM-MLE). This depends on the choice of the fitted correlation through the value distribution. Since the Gumbel (GUM-MLE) distribution rather represents a "worst case" for the respective return period and Bocchiola et al. (2008) confirms the application of this method, these values were chosen for the fracture depth determination. To account for elevation of the release area, a correction factor has to be applied. Blanchet et al. (2009) estimated this factor to be about +/- 2 cm per 100 elevation meters, whereas Swiss practitioners usually use a gradient of 5 cm per 100 elevation meters (Margreth et al., 2008). In consideration of the large avalanche scenarios, we applied 5 cm per 100 elevation meters to each release polygon corresponding to existing studies in the Swiss Alps (Bühler et al., 2022). This elevation correction was applied based on the reference elevation of the measuring station ME2 at 1320m a.s.l. This correction was calculated to each individual release area depending on the difference of the respective altitude compared to the altitude of the measuring station. If, for instance, a release area is 500m higher than the reference altitude, a correction surcharge of 25cm snow would be added to the release area. As it is standard in avalanche practise, for steeper terrain, we assume that less snow accumulates than in flat terrain. Depending on the average inclination of the individual release area, the slope correction factor of the snow height for each PRA is individually calculated and corrected. This is applied automatically in the ArcGIS Python script for fracture depth $d0$ calculation by using a slope factor $\psi$ generally depending on the snow resistance. $\psi = 0.291/sin(\alpha) - 0.202 \times cos(\alpha)$ with $\alpha$ being the slope angle (Salm et al., 1990; Gruber and Margreth, 2001; Bühler et al., 2018) and the fracture depth $d0$ being a function of the 3-day snow depth increase $d0^*$ and the slope factor $\psi$, expressed as $d0 = d0^* f(\psi)$ (Salm et al., 1990).

Snow depth increase data was taken from the SLF-station Meien ME2. This station has a record of 66 years and includes extreme snowfalls (e.g. winter 1950/51 and 1998/99), making the data basis more reliable for extreme events. It is located in the center of the project area at an altitude of 1320 m a.s.l., which is close to the average altitude of the release areas. Since the 3- day snow depth increase (see Fig. 3) at the meteorological station Meien is taken in flat field, a standard slope inclination

**Table 1.** Δ 3-day snow depth accumulation [cm] at the weather station Meien (1320 m a.s.l.) in canton of Uri.

| Scenario | Return Period | GEV-MLE [cm] | GUM-MLE [cm] |
|----------|---------------|--------------|--------------|
| [A] | 30-year | 112 | 114 |
| [B] | 100-year | 130 | 136 |
| [C] | 300-year | 147 | 156 |

correction (Salm et al., 1990; Margreth, 2007) was conducted at all release polygons to consider an adapted fracture depth $d0$. To correct for the influence of drifting snow, a snow drift factor was added depending on the size of the scenario (Salm et al., 1990). In practice, this factor strongly depends on local conditions at the release zones (Margreth, 2007). For the 30-year return period scenario we added 30 cm and for the 100-year return period scenario 40 cm and the 300-year return period scenario 50 cm of drifting snow correction. After defining the fracture depth and the area of the release zones as well as applying the corresponding correction factors, the volume for each release area could be determined and used for the hazard simulation with the RAMMS:LSHIM large scale hazard indication mapping model (Bühler et al., 2018, 2022).

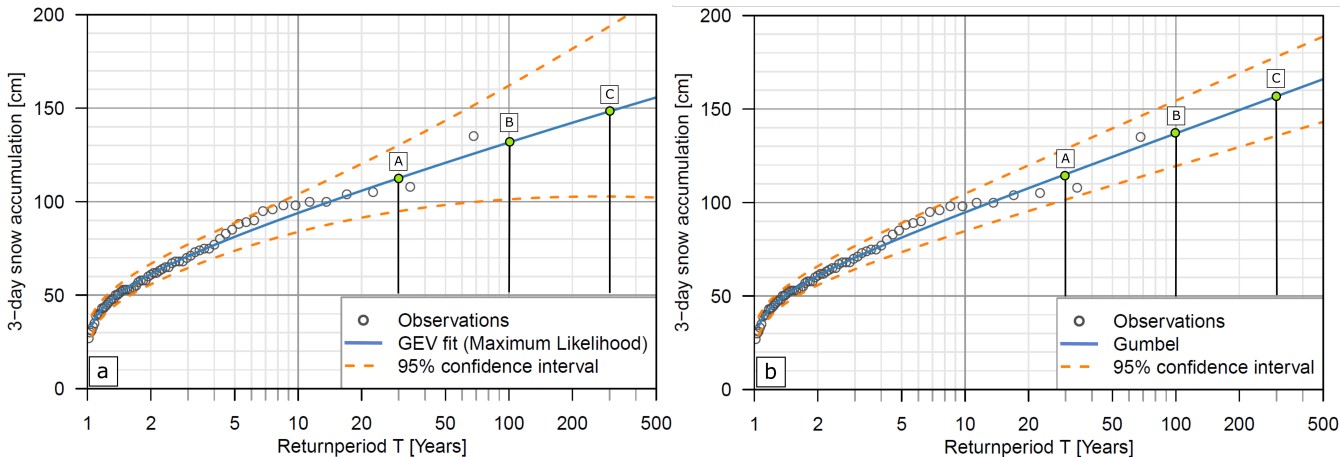

**Figure 3.** Extreme value statistics of the 3-day snow depth increase data (ΔHS3) of the meteo-station Meien (1320 m a.s.l.) for determining scenario return periods, Fig. 3(a): Generalized extreme value distribution with maximum likelihood estimations (GEV-MLE) for return period scenarios [A] = 30-year, [B] = 100-year, [C] = 300-year; Fig. 3(b): Gumbel (GUM-MLE) distribution with maximum likelihood estimations for return period scenarios [A] = 30-year, [B] = 100-year, [C] = 300-year, see Tab. 1.

### 2.3.2 Creation of the avalanche protection forest layer and delineation of the release areas

Forest influences the snowpack structure by interception and changes in the micro-climate. A increased topographic ground surface roughness by forests, and interregular layers in the snowpack prevent avalanche formation. In some cases dense forests are even able to stop movement of small avalanches due to higher friction in the flow- and the avalanche run-out zone (Schnee-

beli and Bebi, 2004; Bebi et al., 2009; Teich et al., 2012; Brožová et al., 2021). To take forest cover into account for hazard mapping, we used the algorithm developed by Bebi et al. (2021) and introduced by Bühler et al. (2022) to identify protective forest in the study area (Fig. 4). The algorithm of Bebi et al. is based on a database of 150 forest avalanches and yields a logistic regression model taking into account the parameters slope gradient, degree of forest cover and gap widths. Together with a vegetation height model and a high resolution elevation model, a generated logistic regression model was used to calculate the avalanche disposition (Bebi et al., 2021). The so-derived protective forest was calculated for frequent (return periods < 100 years) and extreme scenarios (return period > 100 years). The protection forest model is improved and extended by "shrub forest" (= additional bush forest) layer (Weber et al., 2020) and a ground roughness layer. Since forest classified as shrub forest has a reduced protective capacity against avalanche release, the existing protective forest layer was expanded with a shrub forest component to take account of the reduced protective effect for avalanche release. This is a significant improvement over previous protective forest layers in which shrub forest was not specially designated. In this study, in a frequent scenario with a 30-year return period, shrub forest prevents avalanche release, in the large scenarios with 100-year or 300-year return periods, this effect no longer has any influence. The ground roughness layer classifies rough ground to mitigate avalanche release to a certain extent. The so-implemented ground roughness is taken into account for the protective forest layer calculation in frequent avalanche scenarios. This procedure results in a binary protection forest layer (green forest layer in Fig. 4) that divides the area of investigation into areas with and areas without protection forest (Bebi et al., 2021). With a forest layer generated this way, it is assumed for the large-scale avalanche mapping process that an avalanche release in the forest is not possible (Bühler et al., 2018). For more details on the algorithm see the study of Bebi et al. (2021). Further, the formation of avalanches is also altered by infrastructure, such as buildings, railways and roads, which subdivide avalanche release areas into smaller sub-areas. Therefore, they are also added to the binary layer as areas that prevent avalanche release.

For avalanche release volumes < 25'000 m$^3$, the binary forest layer is considered for the flow simulations with assigned friction parameters mu (basal friction) and xi (turbulent friction). Values of mu = 0.020 and xi = 400 were assigned for the simulations in RAMMS::LSHIM, because protective forest has an strong influence on the friction during the run out of the avalanches in frequent scenarios with smaller avalanche volumes. For avalanche release areas with volumes > 25'000 m$^3$, the influence of the protective forest is neglected, since the energy generated during the flow process mostly destroys the protective forest and has often hardly any noticeable effect on the size of the affected area in the run out zone (Bartelt et al., 2017). In the extreme scenarios the potential release zones with avalanche volumes < 25'000 m$^3$ are simulated with forest influence. In the frequent scenarios, on the other hand, the potential release areas with avalanche volumes > 25'000 m$^3$ are simulated also without forest.

### 2.3.3 Generation of automated avalanche release areas

To model the avalanche hazard, it is essential to correctly identify the potential avalanche release areas. In practice and for applications on single slopes, this is usually done individually by expert assessment considering geometrical properties of the avalanche prone slopes. This individual selection is no longer efficient for a large-scale application like in this study and was

therefore done with an automated approach introduced by Maggioni and Gruber (2003) and further developed by Bühler et al. (2013); Veitinger et al. (2016); Bühler et al. (2018), applied to large scales by Bühler et al. (2022).

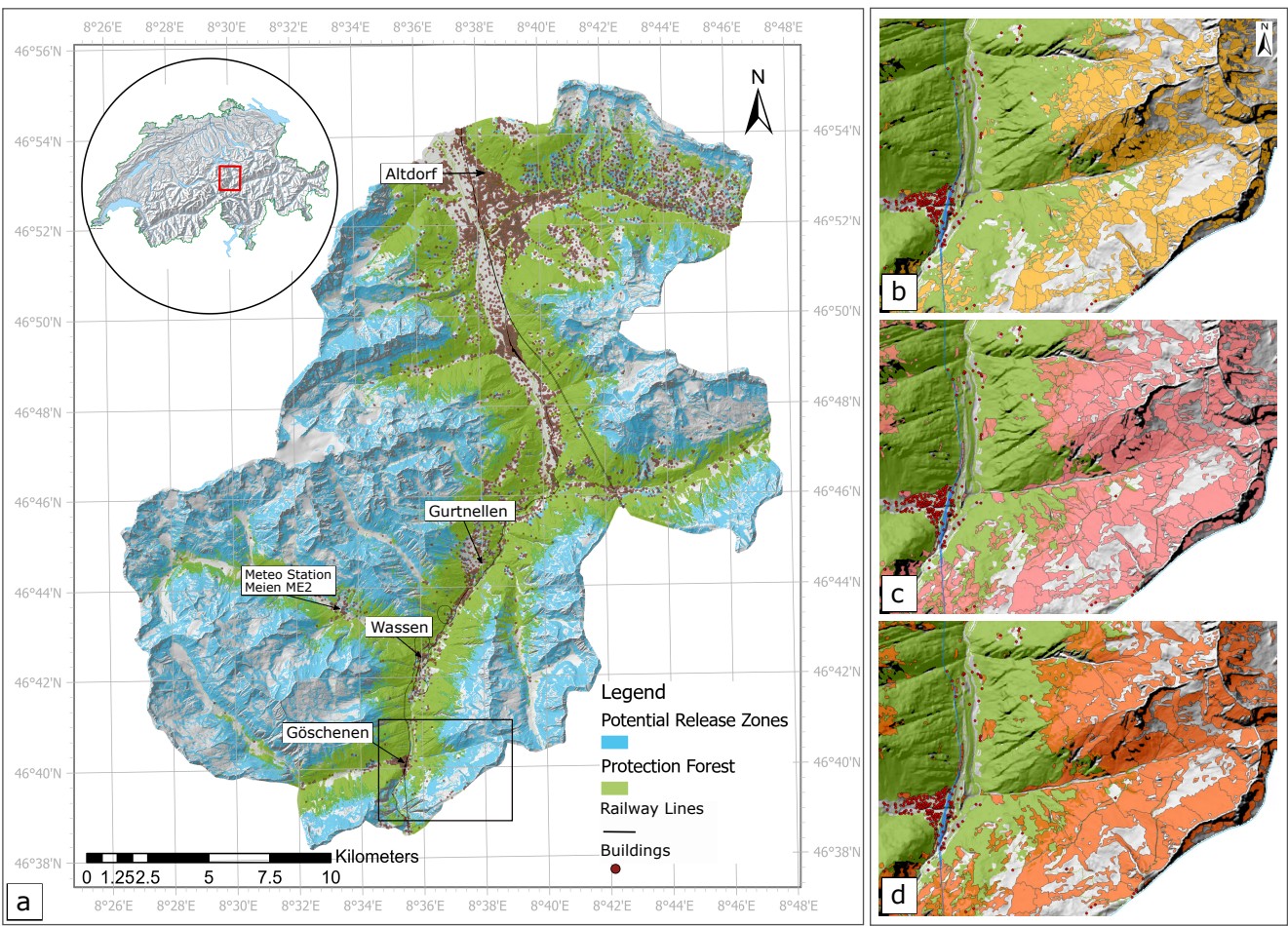

**Figure 4.** (a) Overview of the project area with the automatically generated protection forest layer in green and the area of potential avalanche release in blue, (b-d) automatically generated individual potential avalanche release areas (b) for a 30-year return period, (c) for a 100-year return period, (d) for a 300-year return period. Map base data source: Swiss Federal Office of Topography

This approach is based on a terrain analysis in a GIS and the "object based image classification OBIA" method (Blaschke, 2010) to delineate the individual release polygons, which serve as input for the large scale avalanche hazard indication modelling (Bühler et al., 2013; Bühler et al., 2018). The algorithm analyses the slope geometry and identifies all possible potential avalanche release zones automatically for a given set of input data. With the automated processing, the single potential avalanche release areas are mapped and classified into four release area size classes: large (release volume > 60,000 m³), medium (25,000 - 60,000 m³), small (5,000 - 25,000 m³) and tiny (< 5,000 m³). Depending on these release volumes, different

friction parameters are assigned to the associated avalanche for the subsequent simulation. These friction parameters mu (basal friction) and xi (turbulent friction) are assigned according to the RAMMS standard modelling procedure (Christen et al., 2010) which is known to be standard for hazard mapping in Switzerland.

### 2.3.4 Large scale avalanche hazard indication mapping with RAMMS::LSHIM

We used RAMMS::LSHIM ("Rapid Alpine Mass Movement Simulation ::Large Scale Hazard Indication Mapping"), which is a special version of the base module of RAMMS::AVALANCHE, a numerical avalanche simulation model capable of simulating avalanches in complex topography (Christen et al., 2010). RAMMS::AVALANCHE is based on an efficient second-order numerical solution of the depth-averaged avalanche dynamics equations and the two-parameter Voellmy model (Voellmy, 1955) and has been calibrated with numerous observed avalanches such as for example at the SLF test site in Vallée la Sionne (community of Arbaz, Valais). Further, we used an extended set of algorithms for automatic release area identification (Bühler et al., 2018; Bühler et al., 2018; Bühler et al., 2022) and the standard friction parameter set, which is commonly used in practice (Christen et al., 2010; Gruber and Margreth, 2001). For a detailed description of the RAMMS::AVALANCHE application in RAMMS:LSHIM, see Bühler et al. (2018).

For the simulations, a three-dimensional digital terrain model, with a spatial resolution of 10 m, the automated generated potential release areas and the avalanche fracture depth (Tab. 1) served as inputs to calculate the flow velocity, flow heights and the resulting impact pressure expressed in kPa of 40,000 single avalanches, for all three scenarios (Bühler et al., 2022). Each avalanche was assigned to a release zone and individually saved as a separate file, which allowed us to allocate each avalanche with its pressure to a return period, which could later be used in a risk analysis. Figure 5 shows the resulting large-scale hazard indication map and in detail, how the individual scenarios were simulated considering protection forest.

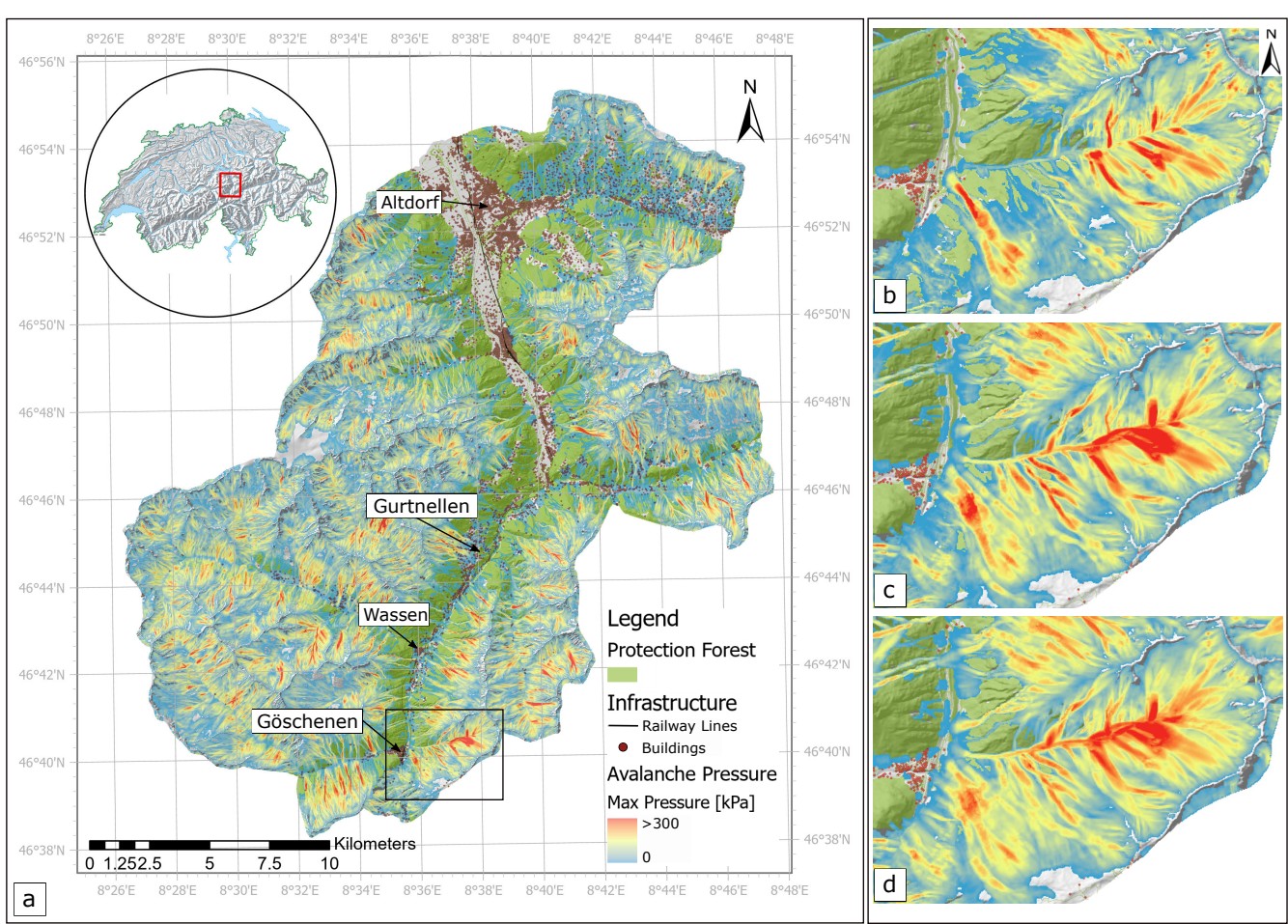

**Figure 5.** (a) Project area with results of the large scale avalanche simulation, pressure in [kPa], (b) detail of the simulation for the frequent scenario, corresponding to a 30-year return period, (c) detail of the simulation for the large scenario, corresponding to a 100-year return period, (d) detail of the simulation for the extreme scenario, corresponding to a 300-year return period. Map base data source: Swiss Federal Office of Topography

## 2.4 Exposure

### 2.4.1 EconoMe

265  EconoMe is an online tool for the evaluation of the effectiveness and economic efficiency of mitigation measures against natural hazards (Bründl et al., 2016). It provides a sophisticated methodology for risk analysis and the evaluation of the cost-benefit-ratio of protective measures and is used by private companies, cantonal and federal authorities for decision support in the subsidies process. We used specific impact functions and building values following the EconoMe methodology as input

values for the risk analysis in this study. All used values for this study, generated by following the EconoMe methodology are described in Sec. 2.4.2 and listed in Tab.2.

### 2.4.2 Generation of high-resolution exposure point data sets with monetary values

Four main data sets were used to determine monetary values: (1) STATPOP, statistical population data which provide information on how many people live in a building; (2) STATENT, statistical data specifying which buildings are used for economic purposes; (3) Construction zones Switzerland, providing information about settlement areas or agricultural zones, and (4) a 3D data set of all Swiss buildings, including the volume of individual objects. To allow a high performance in risk calculation and a broad applicability, buildings in the exposure data frame are considered as point objects located in the center of a building polygon. Using these four data sets, an exposure point layer was created according to the guidelines for creating risk overviews (FOEN, 2020) as described in the following.

The GIS layer with assigned monetary values was created for 13,304 individual objects. To generate this monetary building layer, objects needed to be classified using the above motioned data sets, also listed in Tab. 2. The classification was carried out by plotting the statistical data STATPOP and STATENT on a 2D building layer. The monetary values of buildings were assigned according to the building types also listed in Tab. 2. According to EconoMe standards, the average number of persons in one housing unit (= HU) was assigned as 2.24 persons per unit (FOEN, 2021). By using STATPOP, we identified the number of persons living in each building and determined the number of housing units rounded to the next smaller number. When the number of persons per object was below 2.5, the object was classified as single residential building; when the number of persons per object was larger than 2.5, it was classified as multi-residential building. Based on the number of HU, monetary values could be assigned to the objects as listed in Tab. 2.

Since there is no comprehensive data available for evaluating the value of companies and buildings, the STATENT data set was used. This data provides building coordinates of each registered company in Switzerland. If one or more STATENT data points plot on the location of a building, this building was classified as a business building. The value corresponds to the volume of the building (FOEN, 2021), which was taken from the Swissbuildings3D data set yielding a value of 280 CHF per $\mathrm{m}^3$ volume. For objects with the combined usage "business and living", on which both, the STATENT and STATPOP data points apply, the number of residential units was taken for the assignment of the monetary value, as it provides the higher values. Buildings, located within the Swiss construction zone plan with no inhabitants and a volume of less than $100\,\mathrm{m}^3$, were classified as uninhabited outbuildings with a value of 60,000 CHF per unit (garages and parking lots including vehicles).

Uninhabited outbuildings with a volume greater than $100\,\mathrm{m}^3$ were classified as economically used buildings and their value was determined by the volume. The same applies to buildings that are located outside the Swiss construction zone plan. If their volume is less than $100\,\mathrm{m}^3$ they were classified as agricultural outbuildings with a value of $80\,\mathrm{CHF/m}^3$. If their volume is greater than $100\,\mathrm{m}^3$ they were classified as agricultural main buildings with a value of $180\,\mathrm{CHF/m}^3$ (FOEN, 2020). This method allowed to create a exposure data layer, shown in Fig. 6 with a monetary value for each individual building.

**Table 2.** Overview of the base data used for the creation of the monetary building layer, definition of monetary values of building types; abbreviations: HU = Housing Unit, FSO = Swiss Federal Statistical Office (FSO, 2019), swisstopo = Swiss Federal Office of Topography, ID = EconoMe object identification number.

| Building Type | EconoMe ID | CHF | Data Information | Data Origin |
|---|---|---|---|---|
| Single residential buildings | 1 | 650,000 / HU | Resident population per Building STATPOP | FSO |
| Multi residential buildings | 87 | 550,000 / HU | Resident population per Building STATPOP | FSO |
| Garage and parking | 4 | 60,000 / unit | Construction zones of Switzerland | swisstopo |
| Industrial building | 6 | 280 / m$^3$ | Employed population per Building STATENT | FSO \swisstopo |
| Agricultural use | 2 | 180 / m$^3$ | Swiss BUILDINGS 3D | swisstopo |
| Barn | 3 | 80 / m$^3$ | Swiss BUILDINGS 3D | swisstopo |

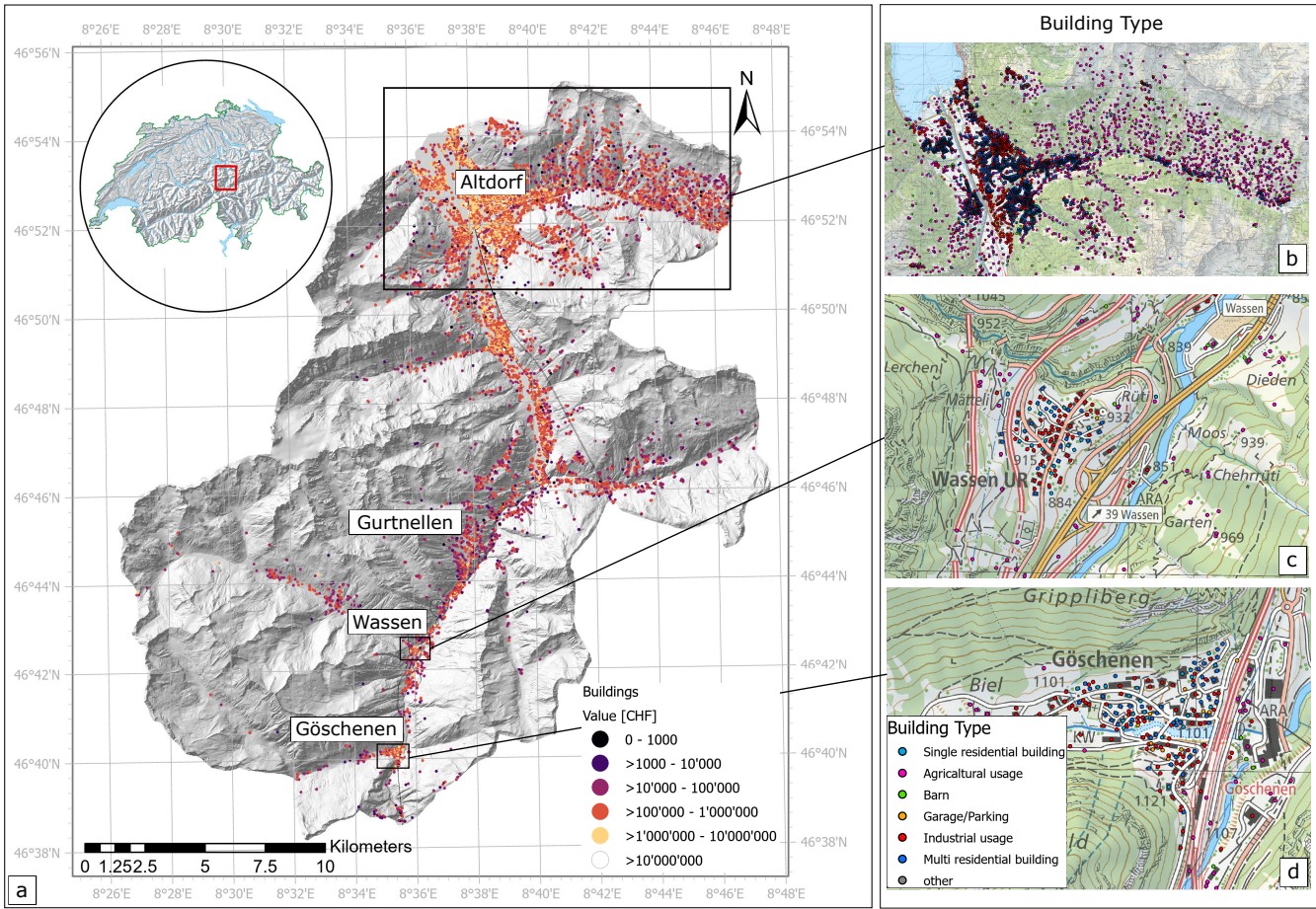

**Figure 6.** (a): Spatial distribution of exposed monetary values in the project area. (b): detail section, different building types matching actual buildings on the map (Map source:Swiss Federal Office of Topography, modified map).

## 2.5 Vulnerability

### 2.5.1 Avalanche impact functions

The extent to which an object is damaged by a hazard is referred to as the "hazard impact" or "impact". We determined the impact by avalanches using so called "impact functions", which is equivalent to "vulnerability function" or "damage function", as defined in the CLIMADA methodology (Aznar-Siguan and Bresch, 2019), which express the damage an object suffers at a certain avalanche intensity (= pressure in kPa). These functions are based on values in EconoMe (FOEN, 2021). These step functions originate from the standardised hazard mapping procedure in Switzerland where continuous avalanche pressures $p$ are divided into three pressure classes ($0 < p \leq 3$kPa, $3 < p \leq 30$kPa, and p > 30kPa). These step functions follow the Swiss standard (FOEN, 2021) and are based on an evaluation of building damages and expert judgement. According to their type of construction, objects show a different damage susceptibility to avalanche pressure. We defined impact functions with three components for each object type and combined them into a impact function set to describe the following avalanche impacts:

- the mean damage degree (mdd), expressing the mean percentage of damage an object suffers under a certain avalanche impact pressure, expressed in kPa;

- the percentage of affected assets (paa) in the hazard zone that suffer damage (100% in the case of avalanches), and

- the mean damage ratio at a certain pressure (mdr), in this study mdr = mdd.

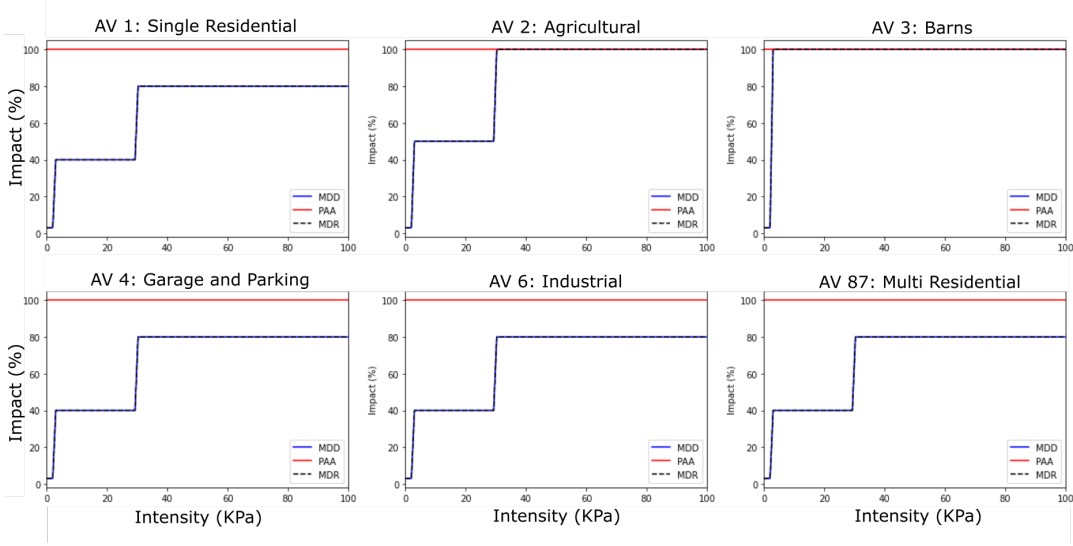

**Figure 7.** Impact functions showing the mean damage degree of different building types exposed to a certain intensity ( = mean avalanche pressure), paa = 100%

The paa describes how many objects suffer damage if exposed to a certain hazard intensity. In this study, we assume that all objects that are affected by a certain avalanche pressure suffer damage (paa is set to 100%). The mean damage ratio (mdr) is a standard vulnerability unit. Explained as the total damage at certain hazard intensity to the affected objects as a ratio of total sums for the all the objects exposed to the hazard. The application of these standard vulnerability components to avalanches is a special case. We assume in this framework that all objects exposed to a certain avalanche pressure also suffer damage according to the mdd thresholds. Therefore, we set the mdr equal to the mdd.

## 2.6 Uncertainty- and Sensitivity analysis

The uncertainty analysis describes the variability of the output of the model given a range of the input parameters, while a sensitivity analysis describes, how a certain uncertainty in the model can be assigned to a certain input parameter (Saltelli et al., 2019; Saltelli, 2002; Kropf et al., 2022). In this study, the model outputs of interest are the average annual impact aggregated (aai agg.) over all exposure points for hazards with different return periods (c.f. Sect. 2.3.1). In general, it is not possible to consider the uncertainty of all input parameters in a model such as CLIMADA (Otth et al., 2022). For this study, we focus on the total exposure value (building value), the mean damage degree thresholds of the impact functions (vulnerability), as well as the uncertainty of the intensity of the avalanches (impact pressure Sect.2.3) as discussed in details in Sect. 2.6.1.

### 2.6.1 Uncertainty analyses

Since the values of the exposure were determined with the generalized value assignment procedure introduced by FOEN (2020), it is clear that these can deviate from real life building values. The method of FOEN provides an estimation of a building value at a large scale, but cannot resolve the differences in value of similar building types. To account for the potentially large fluctuation in asset value, an uncertainty of ± 20% of the total exposures value is assumed. In other words, for each run of the uncertainty sampling, all the assets value are multiplied with a value "et" uniformly sampled from [0.8, 1.2].

Industrial and residential buildings are the most costly building types. The steps of the impact functions for residential buildings and industrial differ by 40%. To cover a wide range of uncertainties, this value of the step jump in the function was taken as the possible uncertainty range. For agricultural buildings or outbuildings, this value can be even higher, as they often have a relatively simple construction. But due to their lower value they play a less important role. To define a "standard case" in our analysis, we consider the function steps of industrial or residential buildings as the variability range. Therefore, for the mean damage degree of the impact function a value range of ± 40% was taken for the uncertainty analyses (c.f. Tab. 3).

To define a range for the hazard intensity (hi) uncertainty analysis, we refer to data recorded in experiments at the Vallée de la Sionne test site. Sovilla et al. (2008a, b) have measured the impact pressure of dry, wet or mixed avalanches, which correspond to different flow regimes. However, in our simulations, we only simulated "dry avalanches", which have mostly higher velocities than 10 m/s. Pressure values of avalanches measured by Sovilla et al., show avalanche pressures in a range between 200 kPa and 700 kPa (Sovilla et al., 2008a). An extrapolation of the data set would yield maximum pressures of about 1000 kPa, which corresponds to the data obtained in our simulations shown in Fig. 5. The average pressure of the

**Table 3.** Uncertainties ranges of the chosen input parameters for the risk assessment

| Parameter | Uncertainty range |
|---|---|
| Exposure (et) | 80% - 120% |
| Mean Damage Degree (mdd) | 60% - 140% |
| Hazard Intensity (hi) | 50% - 150% |

measurements would be around 600 kPa. Therefore, we assumed an uncertainty range of $\pm$ 50% for avalanche pressure (c.f. Tab. 3). The obtained samples are illustrated in the appendix section 5.1 in Fig. 1.

### 2.6.2 Sensitivity analysis

The sensitivity analysis determines which input parameter's uncertainty has the strongest influence on the model output uncertainties (Saltelli et al., 2019). In this study, we used the Sobol sensitivity indices (Sobol, 1993, 2001) as implemented in CLIMADA (Kropf et al., 2022), which can be computed from the same Sobol sequence as used for the uncertainty analysis (c.f. Fig.1, appendix section 5.1). The Sobol sensitivity analysis method is a popular, variance based method that quantifies the contribution of the variance of a single- or two parameters to the unconditional variance of a model output (Nossent et al., 2011). It is expressed as the Sobol sensitivity index. After the distribution of the impact metrics has been computed for all samples (see Sec. 3.2), the sensitivity index for the each metrics was computed. We considered the first order Sobol index S1 that describes the contribution of a single model input to the output variance. This index therefore provides information on the most influential parameter in reference to its system (model).

## 3 Results

### 3.1 Risk maps

With the intersection of hazard maps, the objects at risk and the specific impact functions, we identified the spatial distribution of risk on large scale areas with the CLIMADA risk platform. More precisely, we are able to express risk measures such as the expected annual impact (eai) or the average annual impact (aai) for each individual object at a geographical location taking into account different synthetic hazard scenarios, and translate the damage into monetary terms per year (see Tab. 4). Figure 8 a) - d), shows how different hazard scenarios affect single objects located in the hazardous area. Figure 8 a) depicts the risk pattern of the entire study area. Further risk overviews for the 30-year, 100-year and 300-year return period scenarios can be found in the annex section 5.1 Fig. 2. Figure 8(b) shows a detail of the risk map within a 30-year return period scenario of the Göschenen area and the Rientalbach avalanche path for comparing it with the 100-year in panel c) and the 300-year return period scenario in panel d). The colored dots represent the expected annual impact corresponding to a 30-year return period. The overall aai agg. (average annual impact aggregated) in the entire study area is CHF 5.02 million/year. The detail panel of Fig. 8(c) displays

**Table 4.** Overview of the average damage and the rounded aggregated average annual impacts (aai agg.) for each of avalanche scenarios and all hazard scenarios combined. The rounded aggregated average annual impact corresponds to the overall risk of the entire region.

| Return period [years] | Scenario | Average damage [million CHF] | rounded aai agg. [million CHF/year] |
|---|---|---|---|
| 30 | Small | 150.75 | 5.02 |
| 100 | Medium | 458.60 | 4.59 |
| 300 | Extreme | 947.12 | 3.16 |
| all return periods | Combined | 1390.9 | 9.73 |

the expected annual impact for single objects in an intermediate scenario with an corresponding return period of 100 years. The average annual impact aggregated in the entire region for the 100-year return period scenario is CHF 4.6 million/year. Figure panel 8(c) shows the expected annual impact in the case of an extreme hazard scenario with a corresponding return period of 300 years. The average annual impact calculated for the entire project area in this scenario is CHF 3.2 million/year. The representation of the expected annual impact in the entire project area, over all three hazard scenarios combined is depicted in Fig. 8(d) with an aggregated average annual impact (aai agg.) of CHF 9.73 million/year. In the detail panels of Fig. 8 b) - d) it can be seen that with increasing return period the area affected by avalanches gets larger and the number of objects at risk increases. Nevertheless, the properties in the hazard zone show lower risks in the higher return periods. The aspect of more exposed objects at risk is expressed by the aggregated average annual impact over the entire project area.

As depicted in the risk maps of Fig. 8a) and Fig. 2 of the annex, it is striking that in all scenarios the most densely populated areas (see Fig. 6) in the main valleys do not represent the greatest risk accumulations in the maps. Hot spots are located on the slopes of the main Reuss valley near Wassen a bit further north in Gurtnellen and in the side valleys of Wassen near Meien. In these areas, mountainous terrain with a high number of avalanche flow paths and a denser number of buildings overlap. There, the exposed buildings are mostly agricultural buildings. According to our method, the value of these buildings depend on their volume. Agricultural buildings can be large in volume, but are also very vulnerable to destruction by avalanches (see Fig. 7) due to their construction. This leads to high impacts at these locations, as shown in Fig. 8. However, there are also a few multi-residential buildings in the endangered areas. If these buildings are at high risk, they appear on the map as dark dots. The largest accumulations of buildings at risk are found on the south-east to south-west facing slopes east of Altdorf. The identification of such a hot-spots can play a crucial role in a decision making process as discussed in Sec. 5.

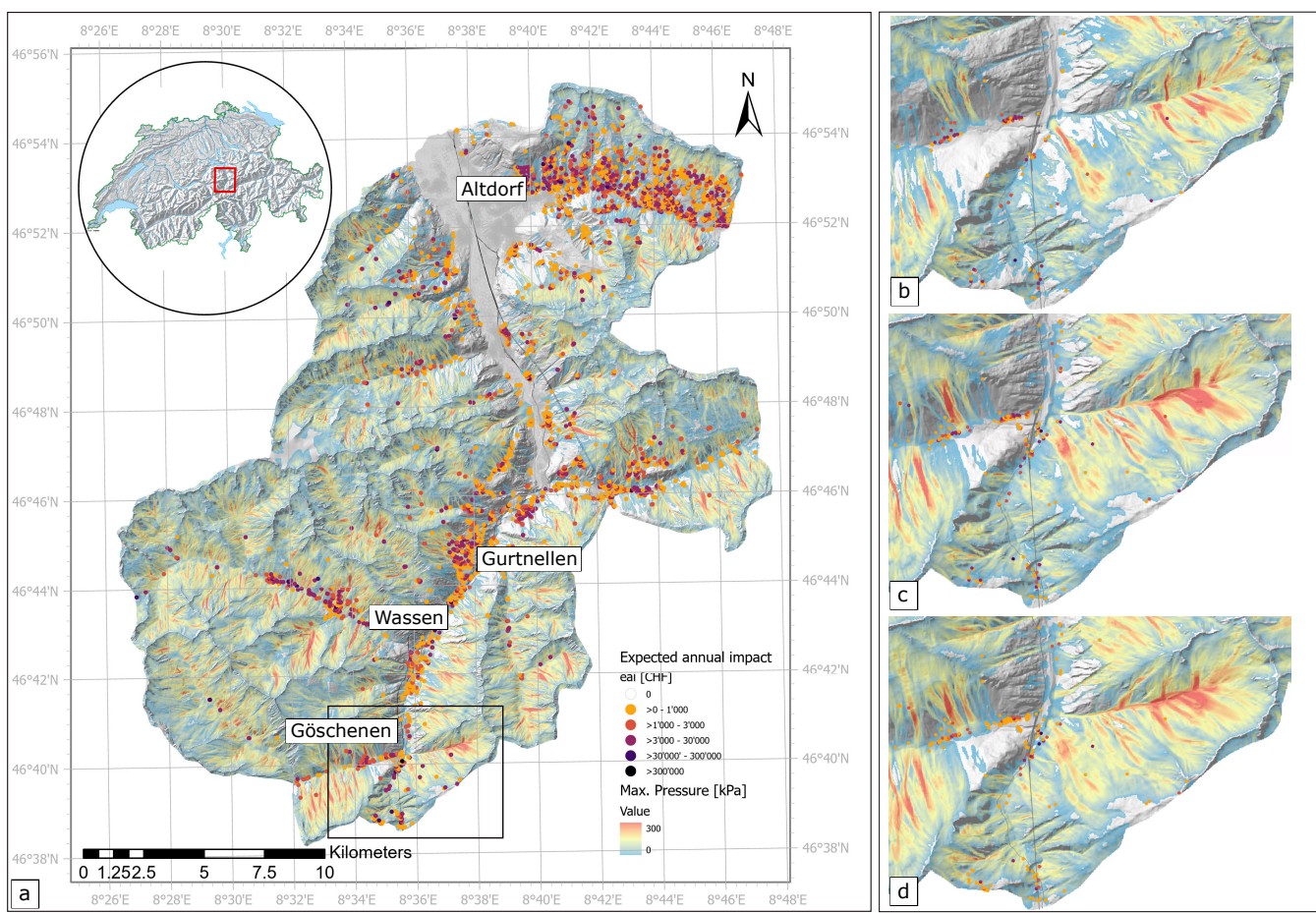

**Figure 8.** Risk maps: Overview of the spatial distribution of expected annual impact for individual objects at the four different avalanche scenarios (a) overview of all hazard scenarios combined (b) detail Göschenen: 30-year return period, (c) detail Göschenen: 100-year return period, (d) detail Göschenen: 300-year return period, Basemap source: Swiss Federal Office of Topography, modified surface model

## 3.2   Impacts and uncertainty ranges and sensitivity analyses

To account for the dispersion of risk under various uncertainties, the annual impact was calculated for randomly sampled cases falling within the defined uncertainty ranges. For the first event set with a 30-year return period (Fig. 9 in blue), the maximum

395   count of samples histogram bars plot between CHF 3.4 million and CHF 5.6 million. This illustrates that the annual impact severity for a hazard with a frequency of 1/30 is likely to lie in this range. The non-aggregated average annual impact values generally range from CHF 2.0 million to CHF 9.0 million. This means a wide dispersion of risk, yet it reflects the broad distribution of the exposed monetary values in the hazard area. Building values ranges from less than CHF 1,000 to maximum building values of CHF 78 million. The sample size was 8,000 generated samples for each parameter and the impact was

calculated for each of these samples. The histogram is not normal distributed, it shows a right skewed distribution with an average impact value of CHF 4.87 million with a standard deviation of CHF 1.34 million. Taking into account all uncertainty ranges discussed above, it is very likely that the annual risk for a 30-year return period scenario, across the entire region will be in the range CHF 3.4 million and CHF 5.6 million per year.

The situation is slightly different, however, for the scenario with a 100-year return period (Fig. 9 in orange) despite higher avalanche pressures, the risk is lower and also with a lower variance. Taking into account the uncertainties, the possible annual impact in the project area is between CHF 2.9 million and CHF 5.4 million. The distribution extends almost over the same range as the 30-year return period scenario (Fig. 9 "Comparison"), but with a sightly higher sample count of 3.5e-7. The average annual impact across all values inside the chosen uncertainty range for the 100-year return period scenario is CHF 4.34 million with a standard deviation of CHF 1.11 million. The histogram for the 100-year return period shows also a right skewed distribution and compared to the 30-year return period distribution, it is almost equally distributed.

Looking at the distribution of the 300-year return period scenario and in the comparison (Fig. 9 in green), we see that the average value of the annual impact is CHF 2.99 million with a standard deviation of 0.73 million CHF; the risk is significantly lower than in the 30-year return period scenario and also lower than in the 100-year return period scenario. The highest sample count is in a variability range of CHF 2.5 million and CHF 3.6 million. The total damage caused by avalanches in the 30-year return period scenario would be approx. CHF 150.8 million, for the 100-year return period scenario, CHF 458.6 million and approximately CHF 947.1 million for the 300-year return period scenario.

In the combined scenario Fig. 10, we see a significantly higher impact range of CHF 7.0 million to CHF 11.5 million with an average value of CHF 9.29 million and a standard deviation of CHF 2.41 million. This shows the great influence of the return period in the calculation of the risk. The risk of a 30-year return period scenario is therefore significantly greater than that of a 300-year scenario.

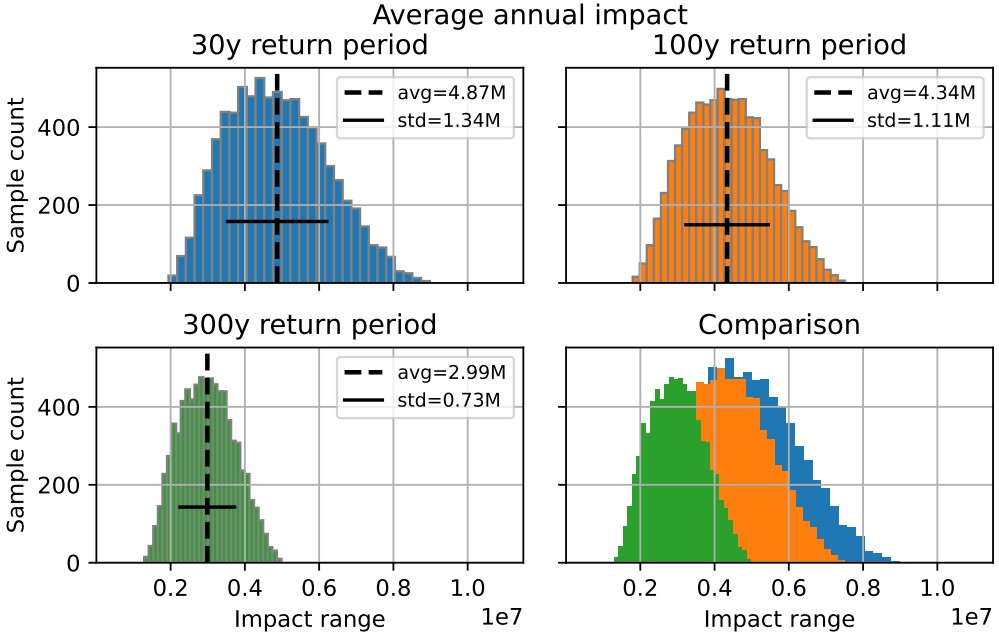

**Figure 9.** Histograms of the calculated distribution of the aggregated average annual impact values from the return period scenarios: blue = 30-year, orange = 100-year, green= 300-year, calculated for eight-thousand randomly pulled samples within defined uncertainty ranges (Tab. 3). avg = average, std = standard deviation

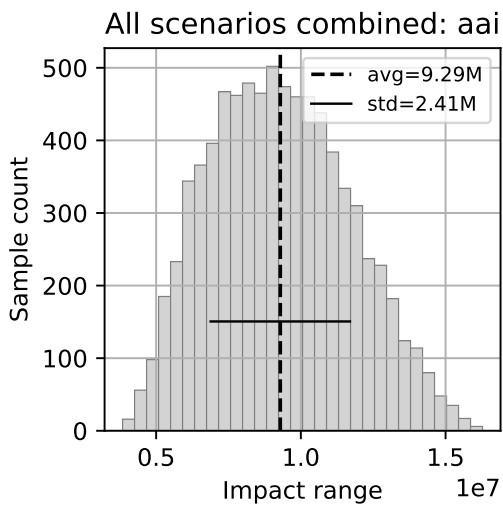

**Figure 10.** Average annual impact aggregated (aai agg.) of all hazard scenarios combined and summed over the entire case study area. Calculated for eight-thousand randomly pulled samples within defined uncertainty ranges (Tab. 3). avg = average, std = standard deviation

### 3.3 Damage frequency curve

A standard in presenting the results of risk analyses are damage frequency curves, which serve to show the calculated impacts and their uncertainty distributions for all considered scenarios. The average damage for the three considered scenarios shown in the risk overviews is depicted in Fig. 11(a). To incorporate a comparison of the designated return period scenarios to the results of the uncertainty analysis, we calculated the 95th and 5th percentile of all damages of the three scenarios of the uncertainty analysis. Thus, for the confidence interval calculation, eight-thousand values were available from the uncertainty analyses for each of the three return periods (30-year, 100-year, 300-year). As shown in Tab. 4 and Fig. 11 the damage for a frequent (30-year return period) scenario is about CHF 150.8 million with a variation from CHF 151 - 216 million (5th - 95th-percentile). In the 100-year return period scenario the average damage is CHF 458.6 million with a variation (5th - 95th-percentile) of about CHF 262 - 625 million. The higher the scenario, the more objects are affected and the larger are the uncertainties expressed in monetary terms. The 300-year scenario shows an average damage of CHF 947.1 million with a large uncertainty range from CHF 548 million - 1.27 billion (5th - 95th-percentile). The dispersion of the uncertainty data for the return periods can be seen in Fig. 11(a) and (b). The median value (black line in the boxplots), calculated from all values of the uncertainty margin of the scenarios is compared with the average damage of the three designated scenarios (see Tab.4 displayed as black square in Fig. 11(b)). Figure 11(b) shows that the average damage values of the three single scenarios are close to the median values of the box plots calculated from all values including the uncertainty margin. It demonstrates that the original calculations for the single damage values (from the three scenarios) are of good quality. Whether the interpolated damages from the scenarios that were not calculated in this study are in a linear relationship to the calculated damage is a rough assumption. Nevertheless, this curve can be used to infer approximate dimensions of the damage to be expected in other return periods, taking into account various uncertainties.

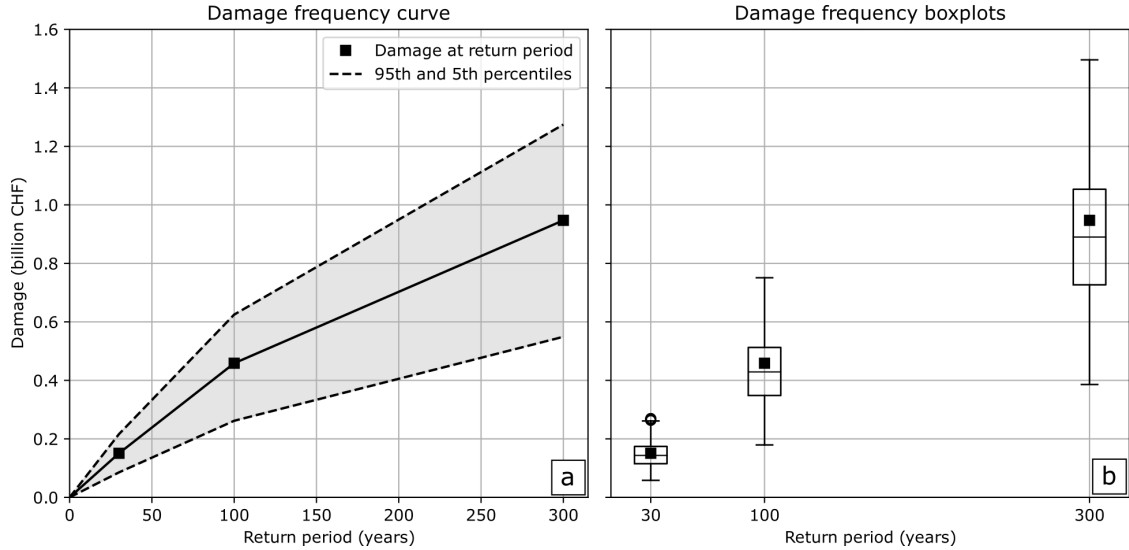

**Figure 11.** (a): Damage-Frequency curve showing the calculated damage at a certain return period with the 95th percentile of all damages calculated within the defined uncertainty ranges, (b): Boxplots of the 3 scenarios depicting all calculated damages of all uncertainty samples (black square = average damage (without return period) of the three scenarios. The boxes extend from [Q1] = first quartile to [Q3] = third quartile of the impacts at return period, the median is depicted as a line. The whisker lines enhance from the box by 1.5 x the inter-quartile range. Points are data exceeding the end of the whiskers.

### 3.3.1 Input parameter sensitivity analysis

The first-order sensitivity index S1 identifies, which parameter impacts the aggregated average annual impact most. Figure 12(a-c) shows that the mean damage degree mdd plays the most important role in this model for all scenarios. For the scenario with a return period of 30 years and the scenario with a return period of 100 years, the exposure "et" and the hazard intensity "hi" seem to be almost equal. In the extreme scenario with a 300-year return period, substantially more objects are affected (see Fig. 8), and the exposure "et" plays a slightly more important role here. The sensitivity index of the hazard intensity in the 30-year and 100-year return period scenario (Fig. 12(a) and 12(b)) is slightly higher than the S1 of the "hi" in the 300-year return period scenario in Fig. 12(c). From this it can be concluded that it is scenario dependent, whether the hazard intensity "hi" or the exposure "et" have more influence. However, in all scenarios the mean damage degree mdd defined by the impact functions is the most relevant parameter. For the combined hazard scenario (Fig. 12(d)) it is similar to the 300-year return period scenario. Both the "et" and the "hi" have almost the same high sensitivity index, but again the mean damage degree (mdd) is more decisive.

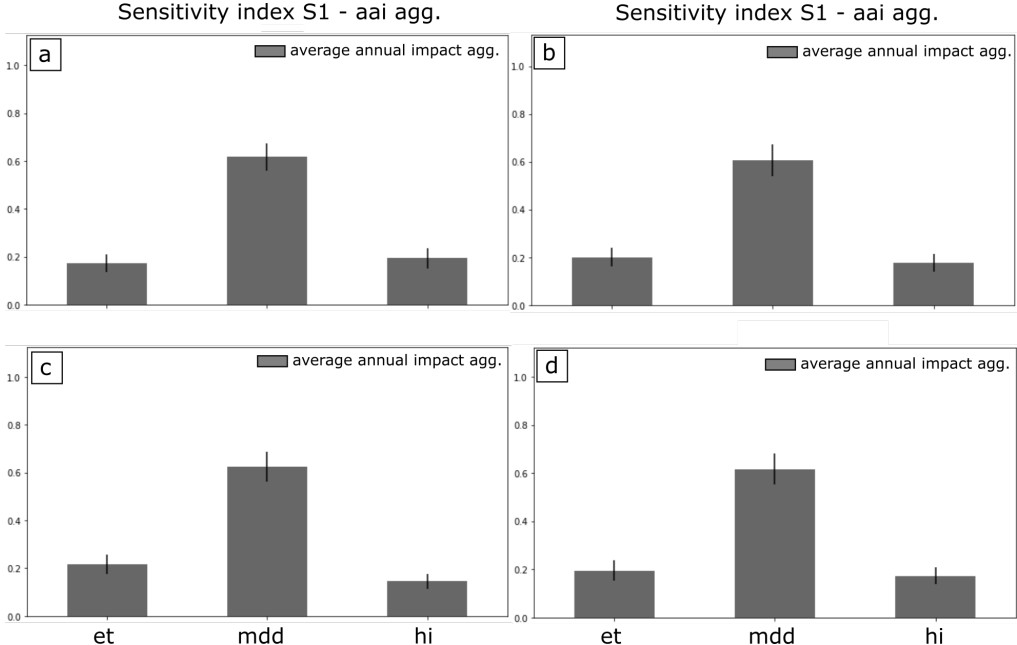

**Figure 12.** S1 - aai agg. = S1 first order sensitivity index and its effect on the average annual aggregated impact value (a) frequent hazard scenario, 30-year return period (b) medium hazard scenario, 100-year return period (c) extreme hazard scenario, 300-year return period (d) over all combined. et = Exposure, mdd = mean damage degree, hi = hazard intensity.

# 4 Discussion

## 4.1 Risk maps and hot spots

As it can be seen in the risk maps of Fig. 8a) and Fig. 2, the densest populated areas (see Fig. 6) in the main valleys are not the areas with high risks. Risk hot spots are in Göschenen and the side valley of Wassen the area around Gurtnellen. They are surrounded by steep slopes with large avalanche release areas in the back country, producing avalanche flows, which can reach down to the valley bottom. The biggest accumulation of avalanche prone object is on the south-west facing slopes east of Altdorf. One explanation for this hot spot is the high number of agricultural buildings and a few residential buildings on very

steep, avalanche prone, open slopes. If one compares risk maps of 8a) and Fig. 2 it can be seen, that depending on the scenario the eai is strongly varying indicated by the different colors, also the numbers of objects affected is different. Even though the number of affected buildings is higher in the large 100-year and 300-year scenarios, the risk of the individual objects (eai) is higher in the 30-year return period scenarios. This can be explained by similar, or slightly higher avalanche pressures at the object, but the lower risks are mostly driven by the influence of the low hazard frequency (= high return period) which

substantially influences the risk. The aggregated annual impact (overall risk) allows a comparison across the entire region and all scenario.

From a risk management perspective, one could conclude that mitigation measures reducing the damage in the 30-year and/or the 100-year return period scenario would be very effective for reducing the overall risk.

## 4.2 Limitations

All of the work steps involved, entail uncertainties and certain limitations. These limitations are addressed and discussed in the following chapters.

### 4.2.1 Large scale avalanche hazard mapping

An avalanche hazard assessment in practice is a multi-step process based on a detailed analysis of the terrain characteristics, local records of snow precipitation data, documented previous events and expertise about the local hazard situation (Rudolf-
Miklau et al., 2014). Avalanche release areas are manually identified by experts performing a detailed terrain assessment including an evaluation of protection forests, which is used as input for a numerical modelling of avalanche run out. In a large-scale avalanche hazard indication mapping, this detailed assessment process is largely omitted, which leads to some limitations (Bühler et al., 2022). Through the use of automated algorithms, potential release areas are only identified on the basis of terrain and scenario characteristics. This is an approximation, as actual avalanche release zones may vary in shape and size in nature.
A validation of this algorithms has been carried out by Bühler and colleagues to evaluate the limitations of the object-based image analysis (OBIA) approach (Bühler et al., 2018). These limitations are compensated by a fast and powerful applicability, which allows to map potential release areas at very large areas, where only the computational capacity and the computation time define the limits of its application. This method is worldwide applicable and requires a small amount of input data, making it one of the unique and most powerful tools for large-scale hazard mapping. We therefore consider that the advantages of the
large-scale applicability and the preciseness of the results correspond very well to the goals of our study.

The automatic calculation of protection forest also entails its limitations. The definition of the protection forest was based on data sets compiled at a specific point in time in the past (2020) using remote sensing methods. With the effects of extreme windstorms or bark beetle infestations or consequences of climate change (e.g. dry periods), the forest structure can change
rapidly (Brožová et al., 2020; Brožová et al., 2021). A detailed consideration of these effects and an expert evaluation of the protective forest (Bebi et al., 2021) is not applicable for large-scale applications and is not imperative for the preparation of risk overviews. This method is based on an evaluated state-of-the-art approach for the definition of avalanche protection forest on large scales, has high quality standards and thus represents an essential basis for the delineation of potential avalanche release areas in our work, and can also be applied to different fields of forest and natural hazard interactions.


Using 3-day snow depth from a single weather station and, based on it, deriving avalanche fracture depth and return period scenarios for an entire region, may result in uncertainties for both, the scenario definition and fracture depth. Both have a direct effect on the avalanche volume and subsequently on the extent of the avalanches in the run out zones. The longer the time series, the more extreme snowfall events would have been recorded. However, a time series of 66 years used in our study

is common practice. The time series available are comparatively long and longer series were not available either. However, this involves bigger uncertainties for estimation of fracture depths at high return periods. As can be seen in Fig. 3, the GEV method results in +/- 30cm for a 100-year return period and +/- 50cm for a 300-year return period, and the Gumbel (GUM-MLE) method results in +/- 20cm for the 100 return period and +/- 25cm uncertainties for the 300-year return period. Due to the 66-year record period, it can be assumed that the estimate of 3-day snow depth increase for a 30-year scenario contains lower uncertainties (+/- 15 cm). This consideration of uncertainties does not take into account local wind effects that can arise because of topography changes at ridges and the back of mountains that significantly influence snow deposition in the release areas. These effects can lead to a high variability of the snow pack in the release zones. However, these local conditions cannot be taken into account in such large-scale applications. Only a wind load factor (30-50cm) depending on the scenario size was added. Without detailed wind/snow deposition studies, this uncertainty in the modelling process is hard to quantify and has to be accepted.

After all input variables have been determined in the avalanche simulation, the topography and the assigned friction parameters play the most important role in the run out flow of an avalanche. For single slope assessments, different parameters (mu and xi) would be assigned for each change in the flow cross-section (gully, channel or open slope) and thus, more precise results could be obtained. In a large scale simulation, a generalized approach is used that regulates the friction parameters via the avalanche volume and takes the topography in the run out into account, but without manually assessed details (please see (Bühler et al., 2022)). This leads to a less differentiated hazard pattern compared to a single slope approach in the avalanche run out zone. Another effect that includes uncertainties is that all theoretically possible potential release areas are simulated with our model. In reality, a more differentiated picture of avalanches would result for each catchment zone in which some release areas produce an avalanche and others do not. Over an entire catchment area, this leads to an overestimation of avalanche risk because too many objects are affected. This effect is not uniform for each slope aspect or catchment area. As far as we know, there are no studies on the release probability of avalanches, that can be applied to large scales to represent an actual natural avalanche release scenario in a model. The advantage of our approach is, however, that all potential avalanche release areas that have the required geometrical properties for avalanche formation, such as slope, aspect, curvature and smoothness, are actually covered in the simulation and taken into account for risk assessment. Here the need for further research is identified to design probabilistic avalanche sets to cover real life avalanche release scenarios. In our risk analysis, each individual avalanche is treated as an individual, independent event and the resulting damage is averaged out over all events impacting an object.

### 4.2.2 Building impacts and avalanche risk mapping

In this paper, we identify the monetary avalanche risk for buildings at a large scale. We intersect impact functions with defined vulnerabilities of buildings with monetary values and continuous hazard indication maps of avalanche pressure. To do this at a large scale, some generalisations have to be made which leads to certain limitations in the level of detail.

The impact functions define the damage rate in steps and pressure classes. Actual damage does not always depend on these

classes. If an avalanche hits an object, it is important at what angle the building is oriented in relation to the flow direction of the avalanche, what is the specific flow regime as well as how high the avalanche flows (Kyburz et al., 2018) or, whether the building has structural weaknesses (e.g. windows or doors) exposed to the avalanche flow. Even though the impact step function was derived from expert assessments and damage surveys of avalanche incidents, it is unlikely that damage occurs according to these steps or is linear to the avalanche pressure - depending strongly on the individual situation. Even minor damage to the foundations of a house can lead to total damage and reconstruction. In our method, all these details cannot be taken into account. To ensure the performance of our approach, we generalize. We consider buildings as point objects with a generalised vulnerability. When using central points in building polygons, we naturally underestimate the influence of the actual three-dimensional building size in the avalanche flow. Small buildings are therefore equally considered as large buildings. Nevertheless, this approach allows a simple and performative applicability and an efficient large-scale approach but introduces inaccuracies in detail. However, we try to address the framework of these uncertainties by means of an overall uncertainty analysis.

Further, our method of the avalanche impact calculation does not take into account the temporal effects of the hazard. In reality, the first avalanche that destroys an object completely, "prevents" its destruction by further events. After many minor avalanches have occurred in the same catchment area, the probability of a major event at the same location decreases. To cover these effects in a risk tool, a detailed probabilistic study would be needed for each individual catchment area. Local meteorological weather events and time-dependent interactions of individual avalanches would have to be investigated in detail. This would increase the level of detail but significantly weakens the independence of the place of application as well as the large-scale applicability for the identification of avalanche risks.

### 4.3  General insights and strengths of the approach

In this work, we have linked a risk tool with a new method for large scale hazard assessment. We have calculated and presented avalanche risks of over 40,000 single avalanches in different hazard scenarios for more than 13,300 buildings over an area of 469.3 km$^2$. Due to the complexity and the small scale of the hazard process (compared to storms or floods), avalanche risk assessment is often addressed at the local level. Studies previously carried out in this field, such as those of Fuchs et al. (2004) and Fuchs et al. (2006) or more recent studies such as Zgheib et al. (2020) have put their focus on the scale of particular villages and are depending on existing hazard maps. Ettlin et al. (2014) established a risk tool for buildings using RAMMS::Avalanche on single slopes. Large scale approaches as the one of Kazakova et al. (2017) assess risk for 60 buildings, others focus on risks and protection forest development such as Renner and Steger (2021). Fuchs et al. (2015) uses detailed avalanche hazard maps at regional scale and flood hazards at a countrywide scale. Bühler et al. (2019) focus on large scale avalanche mapping from satellite data after high intensity snowfall events, but do not address corresponding risk assessment for objects in an entire region. Several studies and expert appraisals were conducted by other authors in the Gotthard area where our case study is located. Most of them are detailed expert judgments on single slopes to assess avalanche risk for specific avalanche paths or specific objects at risk. Margreth and Ammann (2004) conducted a study on the south side of Lukmanier Pass in the central

part of the Swiss Alps to develop hazard scenarios to describe the impact of avalanches on specific structures such as bridges. Margreth et al. (2003), for instance, conducted an avalanche risk analysis using previous versions of the RAMMS avalanche simulation software. They also conducted a case study in the Gotthard region with the focus on avalanche safety on pass roads, but based on a detailed single slope hazard evaluation (Margreth et al., 2003). While such approaches are of high detail and accuracy, they do not allow for large-scale or site-independent application. In contrast, our method allows for modelling avalanche danger for unlimited sized territories with little time expenditure, without using hazard maps previously provided by local authorities. Exposure values can either be generated with existing data or roughly estimated in CLIMADA via night light assessments if no detailed information on buildings is available (Aznar-Siguan and Bresch, 2019). Nightlight assessment takes the light intensity of satellite images taken of the landscape at night and assigns monetary values to a location depending on the light intensity. This approach allows the method to be applied worldwide even with a limited exposure database available. The RAMMS::LSHIM method and the good adaptability of the risk tool is a significant advantage when applying this method in areas with no or limited hazard or asset information available.

Our novel approach for simulating avalanches and assessing the spatially distributed risk through monetary valuation is valid at large scales with a high degree of detail (single-object-resolution). Owing to the avalanche simulations from the RAMMS:LSHIM method, no external avalanche hazard information is needed. CLIMADA is easy to adapt and already being used for a variety of other hazards such as hurricanes and winter storms, and now avalanches. This concept, in combination with large scale hazard mapping, can easily be used for different hazards such as rockfall, debris flows or landslides. Additionally, other exposure scenarios such as traffic routes or energy supply lines or critical infrastructure could be integrated. It provides an overview on risk to objects within an area threatened by natural hazards, and helps practitioners to identify risk hot spots and previously unidentified hazard locations. Thus, the process presented in this paper serves to pinpoint locations of high risk where focused assessments might be necessary for hazard adaptation and mitigation.

## 5 Conclusions Outlook

This approach can be applied globally over large areas and could become particularly useful for further research, when considering anticipated future changes of risk. A potential application might be, to study hazard changes induced by the effects of climate change and / or risk changes caused by socio-economic shifts. Lastly, it can be applied in the context of a probabilistic options appraisal concerning risk reduction measures and adaptation planning as well as cost-benefit calculations. In a subsequent study, we will apply our approach to investigate the impact of climate change on future avalanche risk and further develop the framework for the application of other gravitational alpine mass movements.

*Code and data availability.*

The potential release areas necessary for the reproduction of the hazard simulation and RAMMS simulations described in this paper are publicly available on ENVIDAT www.envidat.ch, the WSL data portal, together with the final submission.

600 CLIMADA is openly available from GitHub https://github.com/CLIMADA-project/climada_python, B(Aznar-Siguan and Bresch, 2019)) under the GNU GPL license (GNU Operating System, 2007). The documentation is hosted on Read the Docs https://climada-python.readthedocs.io/en/stable/, (Aznar-Siguan and Bresch, 2019)) and includes a link to the interactive tutorial for CLIMADA. CLIMADA is permanently available at the ETH Data Archive https://zenodo.org/record/5555825 ((Kropf et al., 2021)). The script reproducing the main results of this paper will be available at https://github.com/CLIMADA-project/
605 climada_papers after the final submission.

## 5.1 Appendix A1

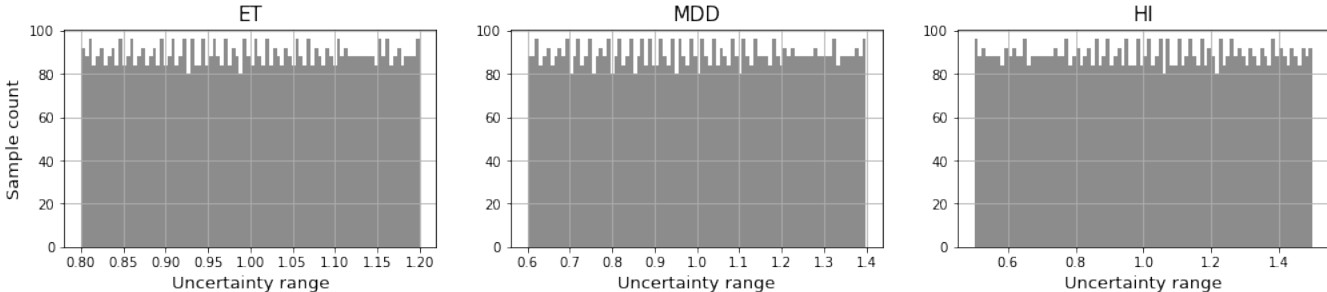

**Figure 1.** Illustration of the automatically generated samples for the chosen uncertainty variables et = Exposure, mdd = Mean Damage Degree (impact function), hi = Hazard Intensity with a sample number of N = 8000.

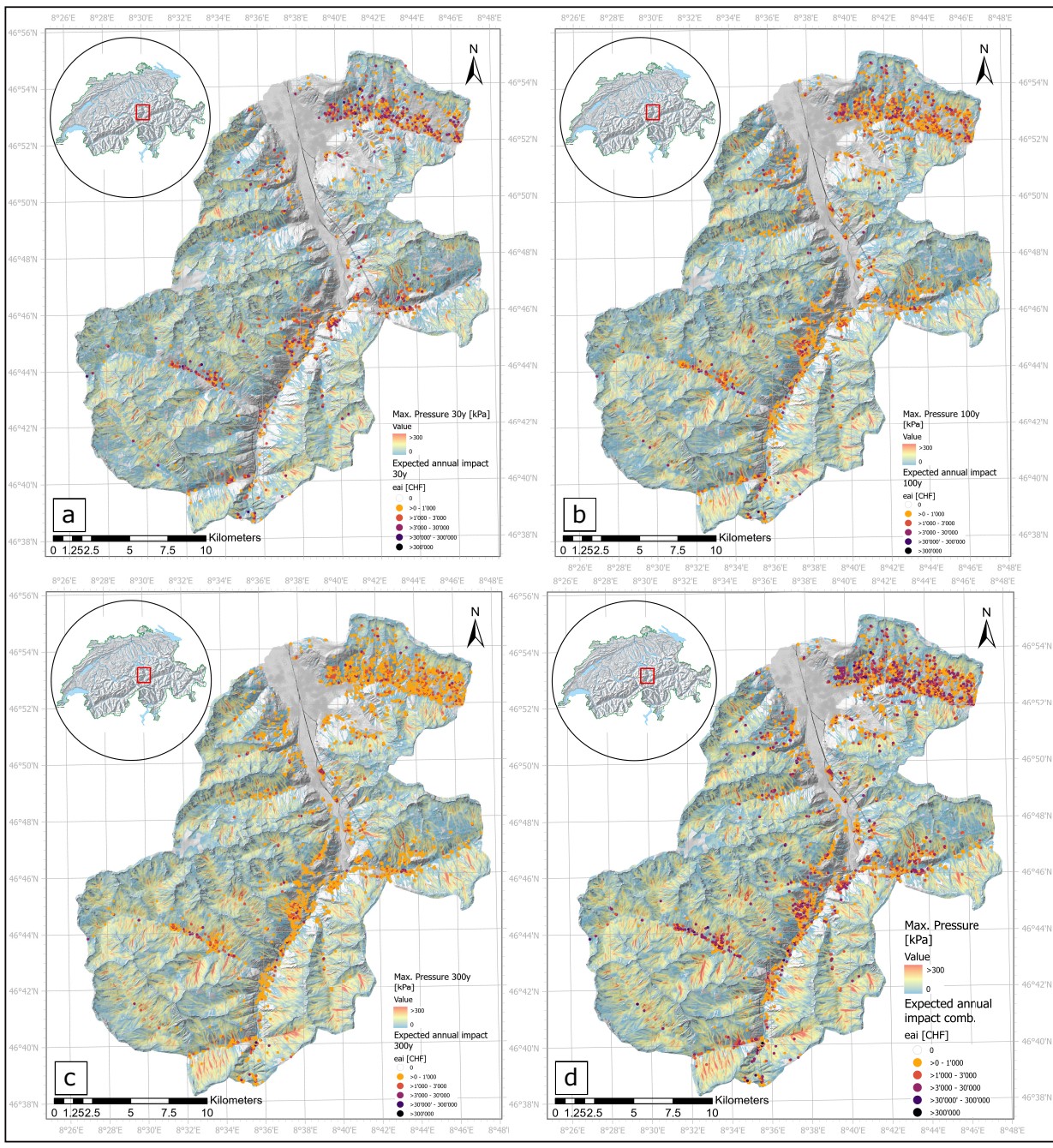

**Figure 2.** Additional Risk maps: Overview of the spatial distribution of expected annual impact for individual objects at the three different avalanche scenarios (a) 30 return period, (b) 100 return period, (c) 300 return period (d) all hazard scenarios combined, Basemap origin (a-d): Swiss federal office of topography: Swisstopo.

*Author contributions.* G.O. and M.B. and C.M.K. designed the study, G.O. and C.M.K. performed the calculations. G.O., D.N.B., T.R. and C.M.K. programmed parts and/or made adaptations of the used risk assessment platform. Y.B. programmed and provided the necessary hazard mapping algorithm G.O. performed the hazard mapping and risk assessment. All authors contributed to the writing process and the validation of the paper, reviewed the results and proposed improvements.

*Competing interests.* The authors have the following competing interests: At least one of the (co-)authors is a member of the editorial board of Natural Hazards and Earth System Sciences. The authors declare that they have no further conflict of interest.

*Acknowledgements.* This work is funded by the WSL research program Climate Change Impacts on Alpine Mass Movements – CCAMM ccamm.slf.ch and the Swiss Railway Company SBB www.sbb.ch/en; we thank both for the financial support. We like to thank the WSL Institute for Snow and Avalanche Research SLF and ETH Zürich for providing their infrastructure and the great work environment, Marc Christen for adjustments to the RAMMS::Avalanche software, Peter Bebi and Gregor Schmucki for providing the forest layer methodology, and Stefan Margreth, Perry Bartelt, Lukas Stoffel, Linda Zaugg-Ettlin, Jan Kleinn, Michael Kyburz, Michael Lehning, Pius Krütli for their opinions and feedback, and Christoph Marty for meteorological data delivery. We further thank Andreas Stoffel for ArcGIS support and Natalie Brozová, Dylan S. Reynolds, Michael T. Lombardo and Amelie Fees for code troubleshooting and scientific exchange.

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
