# Peer review of "Large-scale risk assessment on snow avalanche hazard in alpine regions"

_Natural Hazards and Earth System Sciences, 2022_

## Referee Comment (RC1)

**Review of "Large-scale risk assessment on snow avalanche hazard in alpine regions" by Ortner et al.**

Pascal Haegeli
Simon Fraser University, Vancouver BC, Canada.
August 5, 2022

**Overview**

This paper by Ortner and co-authors describes a new approach for assessing the long-term risk from avalanches to buildings in an entire valley, region or country using a series of existing models. Following the well-established definition of risk = hazard x exposure x vulnerability, the authors use RAMMS::LSHIM to first define probable release areas for three different avalanche scenarios (1/30 yrs, 1/100 yrs, and 1/300 yrs) and then simulate the resulting avalanches to express the hazard in term of impact pressures. They then combine this information with a detailed building layer and building type specific damage functions to estimate the risk in terms of expected annual monetary impact for each building (CHF/yr) and the aggregated average annual monetary impact for the entire study area. The authors also conduct an uncertainty and sensitivity analysis to strengthen their confidence in their approach and explore the impact of uncertainties of individual variables.

The present study is a natural progression of the recent work of the author team, and it explores an important topic that is of broad interest to the NHESS readership because even though the study focuses on avalanches, the model approach is applicable to natural hazard more generally. The presented approach is well thought out and grounded in the existing literature. Overall, I really like the research and the manuscript, but I see two main areas where the manuscript can be improved.

First, the writing is somewhat dense and a bit convoluted at times, which make it difficult for the reader to follow all the details. In addition, tables and figures could be used more strategically, and improving their quality would further strengthen the clarity of the paper. My second main comment is that the discussion is currently very limited almost exclusively focusing on limitations. I believe that expanding the discussion section to better highlight general insight from the study, the strength of the presented model in comparison to existing approaches, the practical implications, and opportunities for future research would make it much stronger. I believe that addressing these two challenges would improve the quality of the manuscript considerably and make it a more accessible and impactful paper. My comments are organized into these two themes.

I hope you find my comments constructive and useful for the revision of you paper.

Pascal Haegeli
Simon Fraser University, Vancouver BC, Canada.

**Manuscript structure, writing and clarity**

**Major comments**

**General**

- This manuscript uses a lot of abbreviations, and not all of them are properly introduced when you use them for the first time. Please be kind to your reader and make it easier for them to find the definition of the various abbreviations. Furthermore, once you have introduced an abbreviation, use it consistently. Also see my comment on Fig. 1 as a potential way for presenting all abbreviations in a central place. I also recommend to only introduce abbreviations when you really need them. Some of them (e.g., GEV-MLE, GUM) might not be necessary since you only use them once or twice in the paper.

**Introduction**

- Line 35: Since the risk definition employed in this paper is broadly used in natural hazards, it would be useful to include a more general reference before you discuss the implementation of the concept in Switzerland.
- Line 45: In my opinion, it would be better to include the description of the overall approach taken in this study (incl. Fig. 1) at the beginning of the methods section and not in the introduction. Instead, the introduction should include a brief overview of the existing approaches that highlights the research need before the objective of this study is defined.
- Line 59: It might be useful to either properly introduce the study area right here or have a dedicated section to do that at the beginning of the methods section. Right now, the description of the study area is somewhat buried in the hazard section, which already describes a specific component of your model. I also recommend that you have a dedicated figure that shows the details the study area without combining it with another aspect of the model. This will help readers unfamiliar with the local geography understand the context of the study area better.
- Fig. 1: While this figure is visually pleasing, it could be made more informative by including a more detailed flowchart that shows how the different datasets and model components work together. This figure could also be used to introduce all variables with their abbreviations. See earlier comment on abbreviations.

**Methods**

- The presentation of the different topics in the hazard section seems a bit convoluted and not follow a linear story line. For example, you already talk about the different scenarios on L100 before you introduce them on L131. You also talk about potential release zones of difference sizes on L114+ before you properly introduce them on L124. I recommend that you reorganize this entire section and present the information in a more linear and logical way. The suggested expansion of Fig. 1 could be part of making this part of your manuscript easier to understand.
- Line 101: It is unclear to me how the forest model is improved and extended with the shrub forest and ground roughness layers. Please explain in more detail.
- Fig. 2: This is the only figure that displays a specific component of the model in detail. In my opinion, this figures in not necessary as this information can be found in Bebi et al. (2021), and

the extension of the model with the shrub forest and ground roughness layers should be described better in the text.

- Line 149: It is unclear to me how exactly the elevation and incline corrections are applied to the probable release areas. Is a single correction factor derived and applied for each entire probable release area or is it done differently?
- Line 185+: This section on the exposure data is extremely convoluted and difficult to follow. One issue is the different datasets, and the repeated references to them. It might be easier for the reader if you describe the three (?) main dataset first before you get into the details of their use. This will reduce the number of required references. In my opinion, much of the relevant derived building information is actually contained in Table 2 and it is not necessary to repeat this information in the text again. So, the text and table should be more complementary.
- Fig. 6: It is unclear to me why the layout of this map figure is very different from the previous figures. I think it would make it easier for the reader the related the content of the different figures to each other if the layout of all the figures with maps were consistent. Furthermore, the purple dots in Panel b are really hard to see. A similar main-panel-and-three-side-panels layout as Fig. 3 and 5 might allow you to zoom into two additional areas of interest (e.g., the SE and SW facing slopes E of Altdorf), which could support the later discussion of the results.
- Line 221+: The definitions of these terms are difficult to understand? Since you are assuming PAA to be 100% for all avalanche pressures and MDR = MDD, I think this could be simplified considerably. It is also unclear to me why the charts show all three variables when only one is used.
- Line 229: It seem to me that CLIMADA should be explained at the beginning of the methods section when you give a general overview of the modelling approach (see earlier comment). Describing these details here is odd since you already referred to CLIMADA when you explained the impact functions.
- Line 247+: This paragraph on uncertainty and sensitivities seems unnecessary as the following section discuss the topics in more detail. Hence this information should be integrated into the subsequent sections. This will make the overall description more concise.
- Fig 8: This figure does not seem to provide useful information beyond what is described in the text. I recommend deleting this figure.
- Table 3: This table does not seem necessary as it does not provide any information beyond what is explained and easily understandable in the text.
- Line 285: A little bit more detail on the Sobol index S1 would help the reader to properly understand what insight it can provide.

**Results**

- Line 291+ and Fig. 9: I think the information presented in Fig 9 is interesting, but you need to describe it in more detail in the text. I also wonder whether the figure would benefit from having the same layout as the previous map figures. It is necessary to show the results from all four scenarios or are the spatial patterns actually quite similar? If possible, the figure could be made more impactful by only showing one scenario, and have three subpanels that zoom in to special areas of interest in the same layout as the previous map figures. The same figures for the other scenarios could be included in an appendix or supplementary material.

- Line 291+ and Fig. 9: The labels of local towns also need to be improved as your description that the hot spots are located on the slopes of the main Reuss valley near Wassen, Gurtnellen, and in the side valley near Meien will not make any sense to readers unfamiliar with the detailed geography of the Reuss Valley. See earlier comment about a map that just shows the study area.
- Line 301: The description of how the combined average damages are calculated belongs into the methods section and not the results.
- Fig 10: This figure is very useful for providing the reader with a sense of the results of the uncertainty analysis, but I think it could be even more insightful if the axis were the same in all panels. This would give the reader a better sense of the locations and spread of these curves, which would further support the discussion on the observed pattern. Since you are presenting the data in bars, it might be easier to have your y-axis in counts or proportions and not as a density. This relates to another comment on the description of counts on L333.
- Fig 11: It is unclear to me why both panels are necessary for this figure. If I understand the presented information correctly, both panels present the same information, in the left panel with the mean and the 5th and 95th percentiles, and in the right panel with the median, the quartiles, the whiskers, and the mean. In my opinion, the left panel is completely sufficient to present the information. The information presented in Fig. 10 could potentially be presented in the same format (mean and percentiles), which would lead to more consistency in the information presentation.
- Sections 3.2 and 3.3: The way I understand these two sections, they both use the results of the uncertainty analysis to provide additional insight into the calculated values for the annual impact severity (Section 3.2) and the aggregated average annual impact (Section 3.3). While the section titles currently focus on the presentation format (uncertainty ranges and damage frequency curves), both of these formats can be applied to either impact measures, and there is considerable overlap between them (e.g., the spread presented in Fig. 11 are uncertainty ranges). Hence it might make sense to reorganize this section and present the results of uncertainty analysis more generally with respect to the two different impact measures.

**Discussion**
- See more detailed comment on discussion below.
- Line 431: Your discussion of "all geometrically possible release areas" is unclear to me. Please explain this in more detail.
- Line 453: The fact that you consider buildings as point objects seems to be an important piece of information that should be mentioned in the methods section. Please move that sentence into the method section.

**Conclusion**
- Line 480: You mention that your approach could also be applied to other natural hazards. To make this point more strongly, I think it would be useful to illustrate it with one or two potential examples. This will be how non-avalanche NHESS readers will connect to your study.

**Minor technical comments**

**General**
- At the beginning of the manuscript, the references to figures and tables are a bit challenging because it is out of order (Fig. 1 on L44, Fig. 5 on L84, Fig. 3 on L86, Fig. 2 on L96). This is

confusing and forces the reader to flip back and forth through the manuscript. Please present figures in the proper order. Also see my comment about a dedicated figure for presenting the study area, which might address this issue.

- There are many in-text citations that are not properly formatted. Some of them are missing parentheses while others have too many.
- Many sentences start with 'In order to'. This can be simplified to just 'To'.

**Introduction**

- Line 21+: You seem to use the term alpine in different ways. For clarity, I recommend using 'mountainous regions' instead of 'alpine landscapes' and 'counties situated in the European Alps' instead of 'alpine countries'.
- Line 24: It is a bit odd that you start the description of the serious winter seasons with the winter 2018/19, but only describe it with a single sentence. If the 2017/18 winter is more insightful, I would start with that winter instead.
- Line 49: At this point, it is not clear why the equation for severity is written in two different ways, and there is no supportive explanation in the text. Why it is done like this becomes clear later in the manuscript, but it is unclear here. Hence, this detail might not be necessary here.
- Line 57: Why limit yourself to adaptation measures? It might be better to talk about avalanche risk management in general, since it includes both mitigation and adaptation.

**Methods**

- Line 127: Is it necessary to refer to scenarios in this sentence before they are properly introduced in the next section?
- Line 134: "The definition of the scenarios is operationalized…" or "… is implemented …" might be a better wording than "… correspond …".
- Line 145: The use of the abbreviation GUM seems unnecessary.
- Line 150: Do you mean "existing studies" instead of "further studies"?
- Table 1: Use the same terminology to describe the 3-day snow depth increase in the caption as in the text. It helps the reader if you use consistent terminology. Also, the square brackets in the scenario column are not necessary.
- Line 164: Why does the subheading say RAMMS::AVALANCHE and not RAMMS::LSHIM?
- Line 183: "Subsidization" is not a very common word. "… to assist in their decisions on government subsidies." might flow easier.
- Line 210: The last sentence in this paragraph is not necessary since you explain the impact function in detail in the next section.
- Table 2: Is the EconoMe ID relevant information for the reader of this paper? I think this column could probably be deleted.
- Line 240+: Why are the abbreviations for these terms lower case? This is different from most other abbreviations. See earlier comment on abbreviations.
- Line 270: In this context, "less important" is a better term than "subordinate".

**Results**

- Table 4: Wouldn't it make more sense to have the return period in the second column or integrated into the first column because it is how the scenarios were defined?

- Line 319: This reference to Fig. 9a seems unnecessary. Potentially this is a mistake and should refer to Fig. 10a instead.
- Line 321: I think that adding "The NON-AGGREGATED average annual values …" to this sentence would improve clarity.
- Line 326: I do not completely understand the last sentence of this paragraph. Please clarify.
- Line 329: A reference to Fig. 10b is missing.
- Line 333: The value 3.5e-7 is a density and not a count. See other comment on changing the y-axis in Fig. 10 to counts or proportions.
- Line 338: For consistency, I think it would be better not to change the units for annual impact. Hence, it should be CHF 0.73 million and not 734.06 kCHF.
- Line 338: In scientific writing, the term "significant" should only be used in the context of statistical significance. Use the terms "considerable" or "substantial" instead.
- Line 344: The last sentence of the paragraph is not necessary.
- Line 349: The first sentence of this paragraph is not necessary because you explained this already. See earlier comment on including the description of the calculation of the combined impact in the methods section.
- Line 357: I don't think this reference to Table 4 is necessary. The scenarios are well established by now.
- Line 375: There is no need for this first sentence as this information is described in the method section already.

**Discussion**

- Line 409: It does not seem necessary to describe the derivation of the fracture depth and avalanche scenarios again. This is described in the methods section already.
- Line 420: It is best to avoid shortened forms in scientific writing (e.g., can't, isn't, etc.).
- Line 436: I believe that "avalanche area" should be "avalanche release area" or even "potential release area". Please use consistent terminology throughout the manuscript.
- Line 449: I believe it should say "structural weak points" or "structural weaknesses" but not "structural weakness points".
- Line 470: I do not understand what you mean with "… out of their focus…".

**Discussion**

Your discussion section is currently almost exclusively a description of the potential limitations of your modelling approach. At the end, you provide a discussion of previous studies carried out in this field (L465-476), but it is rather brief and superficial. At the same time, some of the sections included in the result section seem to have more of a discussion character. Examples include the description of the spatial patterns that emerge from the analysis of the expected annual impact for individual objects (i.e., Fig. 9) and the discussion of the decreasing average annual impact with increasing return periods.

I think the discussion section could potentially be strengthen considerably by expanding it and reorganizing the material in the following fashion:

1) Move the discussion-like paragraphs from the results section into the discussion and combine them into a subsection that discusses the generalizable insight from the analysis beyond the study-site specific results.
2) Expand the comparison with existing research in this area to better highlight the strengths of your approach (some of this is currently included in the conclusion section) and how it expands on the previously existing methods.
3) Finish with a slightly tighter discussion of the limitations that highlight future research opportunities.

I think a structure like this would considerably strengthen the discussion section and the scientific contribution of your paper.

---

## Referee Comment (RC2)

**Large-scale risk assessment on snow avalanche hazard in alpine regions**

Ortner, G.[1,2,3], Bründl, M.[1,2], Kropf, C. M.[3,4], Röösli, T.[3,4] and Bresch, D. N.[3,4]

[1] WSL Institute for Snow and Avalanche Research SLF, 7260 Davos Dorf, Switzerland

[2] Climate Change, Extremes and Natural Hazards in Alpine Regions Research Center CERC, 7260 Davos Dorf, Switzerland

[3] Institute for Environmental Decisions, ETH Zurich, Universitätstr. 16, 8092 Zurich, Switzerland

[4] Federal Office of Meteorology and Climatology MeteoSwiss, Operation Center 1, P.O. Box 257, 8058 Zurich-Airport, Switzerland

Journal: *Natural Hazards and Earth System Sciences*
Manuscript id: https://doi.org/10.5194/nhess-2022-112

November 10, 2022

**Review:**

**1. Overview**

The paper by Ortner et al. describes a framework for spatially evaluating risk on a regional/country wide scale. The framework encompasses probability of release combined with hazard assessment, exposure and vulnerability using the risk assessment platform CLIMADA. The hazard part is evaluated using RAMMS:LSHIM with geometrically computed probable release areas for three different prototypical avalanches (1/30 yr, 1/100 yr and 1/300 yr return periods). This allows the authors to express the hazard in terms of approximated impact pressures. Further techniques are used to classify forested areas and adjust the simulation parameters accordingly. The exposure is evaluated using an identification process for building type in order to spatially represent monetary assets. Specific damage functions are then used to estimate the vulnerability of the buildings and the annual monetary impact estimated (using the software EconoMe) for each building. Alongside this, the aggregated annual monetary impact is calculated for the full evaluated area at a regional scale. Further to this, uncertainty and sensitivity analyses were conducted to assess the variability in input parameters.

The study presented explores an interesting topic with far reaching use cases. I believe the work to be of interest to NHESS readers. The study has a clear purpose and combines established techniques in recent literature for a novel framework for quantitative avalanche risk assessment. Due to this I believe the manuscript will be well suited for publication pending major revisions which the authors should address. These revisions fall into two major categories:

1) Technical detail:

There are a few areas of the text where further explanation is required. Specific examples are listed below in the detailed comments section. Due to the nature of the approach combining several established methodologies it is understandable that full explanation of each of the methods is not completely contained in the manuscript and is appropriately linked to citations, however, I feel in some cases it would be beneficial for the reader to get a bit more insight into the methods used so the paper can stand alone.

2) Structure and writing:

There are a lot of examples of typography errors and I feel there are a couple of sections where the structure could be adapted to have established the key concepts/methods prior to the section. The authors should revise the manuscript taking care to catch mistakes and organize the structure to better relay the scientific message of the paper.

Following up on these comments the authors should address, I have provided some more detailed explanation expanding on the examples where I have spotted some technical issues. I hope that the following comments are useful for the revision of your paper.

**2.  Detailed comments**

**2.1  General**

- Throughout the manuscript there are a variety of typography errors that should be addressed. These fall into the following categories:

  - Figures referenced out of order - the current order goes figure 1, 5, 3, 2 . . . . If appropriate figures should be placed in a strategic order such that the reader isn't required to jump around the paper too much as the contents are discussed in the text.

  - Typos.

  - Some citations in text are wrong type - *e.g.* lacking parenthesis.

  - Some more care needed with sentence structure - wrong use of commas, missing words.

  - Structure issues *e.g.* forest section refers to a lot of definitions that haven't been introduced yet and are not linked in the text - *e.g.* could say "as expanded upon in section . . ." - or adjust the structure of the paper such that the forest section comes after the required definitions - for example - the definition of three avalanche scenarios.

  - Excessive abbreviations / acronyms - not all are needed - and inconsistent formatting and use within text - for example - some are not uniformly applied from figure title to the figure caption. Most are all uppercase but some are lower case - most are introduced more than once and then not used - or the definition repeated - or the same definition applied to two different variables (see *e.g.* Line 221). This is distracting to the reader rather than helpful! I recommend using a consistent heuristic for usage throughout and consider carefully in which cases encoding terms in a abbreviation is beneficial.

  - Units should not be in italics - see *e.g.* $m^3$ vs m$^3$.

  - Some references in bibliography are missing information and are not of uniform style.

**2.2  Introduction**

- Line 24: Apparent missing word in sentence "In winter 2019, for example, exceptional snowfall events [occurred?] which caused high damages".

- Equations (1) and (2) - although give a clearer explanation for the risk formulation used - are lacking in descriptive detail. For example $f_{imp}$ is undefined in the text at the equations and not further referenced. It may also be clearer to the reader if the syntax $f_{\text{impact}}$ is used. Risk is repeatedly referred to as a product of hazard, exposure and vulnerability, however, equations suggest this is not the case - rather - it is a function of hazard, exposure vulnerability and probability of consequence. Probability of consequence is also undefined in the text and needs further explanation.

- Line 41: introduces the idea of vulnerability under the IPCC definition with "economic, material or environmental consequences". The paper then utilizes this concept for vulnerability, however, appears to neglect the environmental consequence and focuses on economic and material consequences. While I think this choice does not affect the scientific contribution of this paper it could be expressed clearer if environmental consequences are intended to be included/neglected and why in the risk framework.

- Line 55: sentence unclear - extra comma?

- Figure 1: abbreviations in input key are undefined. Perhaps expand in the caption or link to text.

**2.3  Methods**

- Line 84: Fig 5 referenced out of order.

- Line 91: Sentence 1 of section 2.1.1 currently doesn't make sense - how does "surface roughness" influence the snowpack structure? I believe a different meaning was intended here. Further to this the concept of surface roughness should be expanded upon here - do you mean of the basal topography or of the snowpack

itself? Also which "transition" is unclear - do you mean the transition to decelerating material - or a specific transition in the topography?

- Figure 2: Typo for "treeheight". VHM introduced twice - once as treeheight and once as vegetation height - are these equivalent or does vegetation height also include shrubforest? This should be clarified. Avalanche disposition is also not well explained as to how it is included as a percentage value. This also needs to the be clarified.

- Line 114: Extreme and frequent scenarios not introduced yet - text should link forward to section 2.1.3 or consider restructuring.

- Line 114: Text should also be more clear on definitions and forest inclusion for all eight potential scenarios *e.g.* frequent / extreme - tiny small medium and large and should not leave it ambiguous to the reader.

- Line 141: Typo "statistic"

- Line 153: Function for $\psi$ stated is incorrect - see Salm, Burkard & Gubler (1990) - which states it as

$$f(\alpha) = \frac{0.219}{\sin(\alpha) - 0.202\cos(\alpha)}. \tag{2.1}$$

It should be checked that the correct form is applied in the ArcGIS code also. The function for the fracture depth $d_0$ is then $d_0 = d_0^* f(\alpha)$, which should be further explained.

- Line 170: An overview of the how RAMMS:LSHIM differs from RAMMS:AVALANCHE should be included in this section.

- Line 195: And overview of the EconoMe methodology should be explained.

- Table 2: Typo in caption - swisstopo repeated.

- Line 221: mean degree of damage = to mean percentage of damage?

- Line 223: What is the conceptual difference between mean damage degree and mean damage ratio at a certain pressure?

- Figure 7: General question on the piecewise-step form of the impact functions - do we really expect a 29 kPa avalanche to give 40% impact but 31 kPa to give 80%? I guess my question lies in the idea that while the piecewise plot gives the context for the three prototypical avalanche scenarios wouldn't a smooth function be more realistic where the impact is interpolated between the three key thresholds?

- Line 229: Citation missing parenthesis. Happens several times in this section and section 2.4.2.

**2.4   Results**

- Line 313: Typo "at this locations".

- Table 4: Should explain how the combined scenario is calculated.

- Figure 9: Black dot at Altdorf label leads to confusion - is this a $10^7$ CHF building? If so then aggregated value estimates don't make sense - if not the symbol should be removed. In panel (a) the label also covers points, which should be corrected.

- As a general point about figure 9 - the 1/30 yr expected annual impact seems pessimistic - it would be interesting to see how this compares to actual spending in this area, say from insurance records / payouts or knowing which buildings have been damaged in the last 30 yrs - 50 yrs.

- Figure 10: Typo - "one eight-thousand randomly pulled samples" - doesn't make sense. Also missing units for impact range.

- Line 363: missing CHF?

**2.5 Discussion**

- Line 453: "We consider buildings as point objects ..." - how does this connect with the mapping - do you consider the building as the point at center of the building area/extent? Can this play a role with larger buildings say large barns or warehouses in the studied regions?

- Line 468: "small scale of hazard process" - this is a bit ambiguous. Please clarify the scale intended.

- Line 473: "night light assessments" is not a common term, please clarify.

**2.6 Conclusion**

- Line 482: "previously unknown threats"... - clarification needed.

**Bibliography**

Salm, B, Burkard, André & Gubler, H U 1990 Berechnung von fliesslawinen. Eine Anleitung fuer praktiker mit beispielen. *Tech. Rep.*. Eidg.\ Institut\ für Schnee- und Lawinenforschung, CH–7260 Davos-Weissfluhjoch, publication Title: Mitteilungen des Eidgenössischen Instituts für Schnee und Lawinenforschung Volume: 47.

---

## Author Comment (AC2)

**Final author comments (AC) on : Large-scale risk assessment on snow avalanche hazard in alpine regions**

The final authors comments are indicated in the text by "AC:"

**Referee Comment 1 (RC1) by Pascal Haegeli: Simon Fraser University, Vancouver BC, Canada.**

**August 5, 2022**

**Overview**

This paper by Ortner and co-authors describes a new approach for assessing the long-term risk from avalanches to buildings in an entire valley, region or country using a series of existing models. Following the well-established definition of risk = hazard x exposure x vulnerability, the authors use RAMMS::LSHIM to first define probable release areas for three different avalanche scenarios (1/30 yrs, 1/100 yrs, and 1/300 yrs) and then simulate the resulting avalanches to express the hazard in term of impact pressures. They then combine this information with a detailed building layer and building type specific damage functions to estimate the risk in terms of expected annual monetary impact for each building (CHF/yr) and the aggregated average annual monetary impact for the entire study area. The authors also conduct an uncertainty and sensitivity analysis to strengthen their confidence in their approach and explore the impact of uncertainties of individual variables. The present study is a natural progression of the recent work of the author team, and it explores an important topic that is of broad interest to the NHESS readership because even though the study focuses on avalanches, the model approach is applicable to natural hazard more generally. The presented approach is well thought out and grounded in the existing literature. Overall, I really like the research and the manuscript, but I see two main areas where the manuscript can be improved.

First, the writing is somewhat dense and a bit convoluted at times, which make it difficult for the reader to follow all the details. In addition, tables and figures could be used more strategically, and improving their quality would further strengthen the clarity of the paper. My second main comment is that the discussion is currently very limited almost exclusively focusing on limitations. I believe that expanding the discussion section to better highlight general insight from the study, the strength of the presented model in comparison to existing approaches, the practical implications, and opportunities for future research would make it much stronger. I believe that addressing these two challenges would improve the quality of the manuscript considerably and make it a more accessible and impactful paper. My comments are organized into these two themes.

I hope you find my comments constructive and useful for the revision of your paper.

**Manuscript structure, writing and clarity Major comments**

General

- This manuscript uses a lot of abbreviations, and not all of them are properly introduced when you use them for the first time. Please be kind to your reader and make it easier for them to find the definition of the various abbreviations. Furthermore, once you have introduced an abbreviation, use it consistently. Also see my comment on Fig. 1 as a potential way for presenting all abbreviations in a central place. I also recommend to only introduce abbreviations when you really need them. Some of them (e.g., GEV-MLE, GUM) might not be necessary since you only use them once or twice in the paper.

AC: We will introduce abbreviations earlier and in a clearer manner, following the recommendation of both referees

**Introduction**

 Line 35: Since the risk definition employed in this paper is broadly used in natural hazards, it would be useful to include a more general reference before you discuss the implementation of the concept in Switzerland. AC: We will start with the IPCC definition first and later discuss the implementation of the concept in Switzerland

- Line 45: In my opinion, it would be better to include the description of the overall approach taken in this study (incl. Fig. 1) at the beginning of the methods section and not in the introduction. Instead, the introduction should include a brief overview of the existing approaches that highlights the research need before the objective of this study is defined.

AC: We will follow your suggestion and include the description of the overall approach at the beginning of the method section..

Line 59: It might be useful to either properly introduce the study area right here or have a dedicated section to do that at the beginning of the methods section. Right now, the description of the study area is somewhat buried in the hazard section, which already describes a specific component of your model. I also recommend that you have a dedicated figure that shows the details the study area without combining it with another aspect of the model. This will help readers unfamiliar with the local geography understand the context of the study area better.

AC: We have attempted to keep the number of figures to a minimum and have therefore decided on a combination. But we are happy to add one more figure and remove figure 2 as discussed later. We will adapt the introduction of the study site.

 Fig. 1: While this figure is visually pleasing, it could be made more informative by including a more detailed flowchart that shows how the different datasets and model components work together. This figure could also be used to introduce all variables with their abbreviations. See earlier comment on abbreviations.

AC: We will introduce RAMMS:LSHIM and PRA as well as Risk abbreviation in the figure for better clarity. We deliberately decided not to use a flowchart to keep the illustration clear and concise. We wanted to show how our individual components form Hazard, Vulnerability and Exposure and how this merge with the IPCC risk concept. For a future paper we would like to extend this figure with climate change, so we want to keep this "base figure" as simple as possible. If there are no too strong objections against it we would like to leave this figure as it is.

**Methods**

The presentation of the different topics in the hazard section seems a bit convoluted and not follow a linear story line. For example, you already talk about the different scenarios on L100 before you introduce them on L131. You also talk about potential release zones of difference sizes on L114+ before you properly introduce them on L124. I recommend that you reorganize this entire section and present the information in a more linear and logical way. The suggested expansion of Fig. 1 could be part of making this part of your manuscript easier to understand.

AC: we agree, and will reorganise this section in a more reader friendly fashion

- Line 101: It is unclear to me how the forest model is improved and extended with the shrub forest and ground roughness layers. Please explain in more detail.

**AC: will be adapted**

- Fig. 2: This is the only figure that displays a specific component of the model in detail. In my opinion, this figures in not necessary as this information can be found in Bebi et al. (2021), and the extension of the model with the shrub forest and ground roughness layers should be described better in the text.
   AC: we will focus on the description in the text body and remove the figure from the manuscript and refer for more detail to Bebi et al. (2021). The shrub forest paragraph will be expanded and explained in more detail.
- Line 149: It is unclear to me how exactly the elevation and incline corrections are applied to the probable release areas. Is a single correction factor derived and applied for each entire probable release area or is it done differently?

**AC: Since Referee 1 and Referee 2 had comments on these parts of the text, we will describe the exact procedure again in more detail as in Buehler et al. 2018, so that the paper can better stand on its own.**

Line 185+: This section on the exposure data is extremely convoluted and difficult to follow. One issue
is the different datasets, and the repeated references to them. It might be easier for the reader if you
describe the three (?) main dataset first before you get into the details of their use. This will reduce
the number of required references. In my opinion, much of the relevant derived building information

is actually contained in Table 2 and it is not necessary to repeat this information in the text again. So, the text and table should be more complementary.

AC: We will introduce the main datasets at the beginning and reorganize this section so that it will be more understandable. We will modify figure 6 in a similar manner as fig. 3 and fig. 5 to make it a bit clearer and give a better overview of the exposure dataset.

Fig. 6: It is unclear to me why the layout of this map figure is very different from the previous figures. I think it would make it easier for the reader the related the content of the different figures to each other if the layout of all the figures with maps were consistent. Furthermore, the purple dots in Panel b are really hard to see. A similar main-panel-and-three-side-panels layout as Fig. 3 and 5 might allow you to zoom into two additional areas of interest (e.g., the SE and SW facing slopes E of Altdorf), which could support the later discussion of the results.

**AC: We agree and will present Fig. 6 in a similar way as Fig. 3 and Fig. 5. This should allow us to show the various components of the building dataset in a more understandable way and zoom more into the details.**

 Line 221+: The definitions of these terms are difficult to understand? Since you are assuming PAA to be 100% for all avalanche pressures and MDR = MDD, I think this could be simplified considerably. It is also unclear to me why the charts show all three variables when only one is used.

AC: For our application we use the standard format of the Climada impact functions as presented in the Climada online documentation. Since for avalanches a special case of these impact functions is used, different components of the standard method are set equal (MDR = MDD because of PAA is 100%). We will explain this equalization better in the text body but keep the Climada standard format of the impact functions.

- Line 229: It seem to me that CLIMADA should be explained at the beginning of the methods section when you give a general overview of the modelling approach (see earlier comment). Describing these details here is odd since you already referred to CLIMADA when you explained the impact functions.

AC: Our idea would have been to explain the individual components step by step in the Methods section (like: hazard, exposure and vulnerability, which represent the risk). But we will reorganise it and the Climada section will follow the introduction of the risk concept early in the methods section. Later hazard, exposure and vulnerability will be explained in detail.

- Line 247+: This paragraph on uncertainty and sensitivities seems unnecessary as the following section discuss the topics in more detail. Hence this information should be integrated into the subsequent sections. This will make the overall description more concise.

AC: agreed and will be implemented as proposed

Fig 8: This figure does not seem to provide useful information beyond what is described in the text. I
recommend deleting this figure.

AC: We have included Fig. 8 in the manuscript for the purpose of comprehensibility. We generally agree and will remove it.

Table 3: This table does not seem necessary as it does not provide any information beyond what is
explained and easily understandable in the text.

AC: For us, this table summarizes what is explained in the text a bit more clearly. therefore, despite slight redundancy, we would like to keep it if there are no too strong objections.

 Line 285: A little bit more detail on the Sobol index S1 would help the reader to properly understand what insight it can provide. \$

AC: we will add a few paragraphs and go more into detail.

**Results**

Line 291+ and Fig. 9: I think the information presented in Fig 9 is interesting, but you need to describe it in more detail in the text. I also wonder whether the figure would benefit from having the same layout as the previous map figures. It is necessary to show the results from all four scenarios or are the spatial patterns actually quite similar? If possible, the figure could be made more impactful by only showing one scenario, and have three subpanels that zoom in to special areas of interest in the same layout as the previous map figures. The same figures for the other scenarios could be included in an appendix or supplementary material.

AC: We agree, we will rearrange this figure by showing only the total risk and two or three subfigures, which zoom into areas of interest. The 30-, 100-, 300-year scenario will be in the annex.

 Line 291+ and Fig. 9: The labels of local towns also need to be improved as your description that the hot spots are located on the slopes of the main Reuss valley near Wassen, Gurtnellen, and in the side valley near Meien will not make any sense to readers unfamiliar with the detailed geography of the Reuss Valley. See earlier comment about a map that just shows the study area.

**AC: Agreed and will be corrected**

- Line 301: The description of how the combined average damages are calculated belongs into the methods section and not the results.

**AC: Will be corrected**

Fig 10: This figure is very useful for providing the reader with a sense of the results of the uncertainty analysis, but I think it could be even more insightful if the axis were the same in all panels. This would give the reader a better sense of the locations and spread of these curves, which would further support the discussion on the observed pattern. Since you are presenting the data in bars, it might be easier to have your y-axis in counts or proportions and not as a density. This relates to another comment on the description of counts on L333.

**AC: Will be corrected**

Fig 11: It is unclear to me why both panels are necessary for this figure. If I understand the presented information correctly, both panels present the same information, in the left panel with the mean and the 5th and 95th percentiles, and in the right panel with the median, the quartiles, the whiskers, and the mean. In my opinion, the left panel is completely sufficient to present the information. The information presented in Fig. 10 could potentially be presented in the same format (mean and percentiles), which would lead to more consistency in the information presentation.

AC: Yes, both panels show similar information. Theleft figure shows all the data from the uncertainty analysis with its 95% and 5% percentiles with the damage frequency curve calculated based on the three scenarios. The figure on the right compares all the data from the uncertainty analysis with the data from the individual scenarios and shows that the damage frequency curve from single scenarios matches the median data from the uncertainty analysis and confirms that the single scenarios produce similar values as the uncertainty analysis considering parameter variety. We will explain this in the text better so that it becomes more clear. Therefore, we suggest to keep both panels as long as there are no strong objections.

Sections 3.2 and 3.3: The way I understand these two sections, they both use the results of the uncertainty analysis to provide additional insight into the calculated values for the annual impact severity (Section 3.2) and the aggregated average annual impact (Section 3.3). While the section titles currently focus on the presentation format (uncertainty ranges and damage frequency curves), both of these formats can be applied to either impact measures, and there is considerable overlap between them (e.g., the spread presented in Fig. 11 are uncertainty ranges). Hence it might make sense to reorganize this section and present the results of uncertainty analysis more generally with respect to the two different impact measures.

AC: We can combine the two chapters and present them in a more concise way to eliminate any overlap.

**Discussion**

- See more detailed comment on discussion below.
- Line 431: Your discussion of "all geometrically possible release areas" is unclear to me. Please explain this in more detail.

**AC: will be corrected**

- Line 453: The fact that you consider buildings as point objects seems to be an important piece of information that should be mentioned in the methods section. Please move that sentence into the method section.

AC: Ok

**Conclusion**

Line 480: You mention that your approach could also be applied to other natural hazards. To make this
point more strongly, I think it would be useful to illustrate it with one or two potential examples. This
will be how non-avalanche NHESS readers will connect to your study.

AC: We will mention that Climada is already used to model the risk of other natural hazards like hurricanes and winter storms. We will mention that our approach of combining large-scale modelling of hazard processes and combining them with exposure and vulnerability data to model risk can also be applied to other mass movements such as debris flow, shallow landslides or rock-fall. Our next step will be to apply this framework to large-scale modelling of rock fall risk.

**Minor technical comments**

General

At the beginning of the manuscript, the references to figures and tables are a bit challenging because it is out of order (Fig. 1 on L44, Fig. 5 on L84, Fig. 3 on L86, Fig. 2 on L96). This is confusing and forces the reader to flip back and forth through the manuscript. Please present figures in the proper order. Also see my comment about a dedicated figure for presenting the study area, which might address this issue.

**AC: since both referees have commented we will correct this**

- There are many in-text citations that are not properly formatted. Some of them are missing parentheses while others have too many.

**AC: will be corrected**

- Many sentences start with 'In order to'. This can be simplified to just 'To'.

AC: Ok

**Introduction**

 Line 21+: You seem to use the term alpine in different ways. For clarity, I recommend using 'mountainous regions' instead of 'alpine landscapes' and 'counties situated in the European Alps' instead of 'alpine countries'.

**AC: Ok**

Line 24: It is a bit odd that you start the description of the serious winter seasons with the winter 2018/19, but only describe it with a single sentence. If the 2017/18 winter is more insightful, I would start with that winter instead.

**AC: Ok**

- Line 49: At this point, it is not clear why the equation for severity is written in two different ways, and there is no supportive explanation in the text. Why it is done like this becomes clear later in the manuscript, but it is unclear here. Hence, this detail might not be necessary here.

AC: Equation taken from Aznar-Siguan and Bresch (2019) will be corrected and explained clearer in the text

- Line 57: Why limit yourself to adaptation measures? It might be better to talk about avalanche risk management in general, since it includes both mitigation and adaptation.

AC: Ok

**Methods**

- Line 127: Is it necessary to refer to scenarios in this sentence before they are properly introduced in the next section?

AC: Its not scenarios its avalanche release volume classes. Avalanches are automatically classified by algorithm in volume categories each with defined RAMMS parameter settings for technical reasons, according to RAMMS::LSIHM procedure of a certain scenario e.g avalanche class "large" in 30y return period, will be explained in more detail.

- Line 134: "The definition of the scenarios is operationalized..." or "... is implemented ..." might be a better wording than "... correspond ...".

AC: Ok

- Line 145: The use of the abbreviation GUM seems unnecessary.

AC: Ok

- Line 150: Do you mean "existing studies" instead of "further studies"?

AC: yes, will be corrected

- Table 1: Use the same terminology to describe the 3-day snow depth increase in the caption as in the text. It helps the reader if you use consistent terminology. Also, the square brackets in the scenario column are not necessary.

AC: Ok

- Line 164: Why does the subheading say RAMMS::AVALANCHE and not RAMMS::LSHIM?
- AC: RAMMS::LSIHM is the large scale application of RAMMS::Avalanche. We will correct the subheading.
- Line 183: "Subsidization" is not a very common word. "... to assist in their decisions on government subsidies." might flow easier.

**AC: Ok**

- Line 210: The last sentence in this paragraph is not necessary since you explain the impact function in detail in the next section.

AC: Ok

- Table 2: Is the EconoMe ID relevant information for the reader of this paper? I think this column could probably be deleted.

AC: With the ID the used building type can be found in the online documentation of EconoMe. If someone is looking at the documentation, it might be a useful information. We would like to leave the column it in for the completeness.

- Line 240+: Why are the abbreviations for these terms lower case? This is different from most other abbreviations. See earlier comment on abbreviations.

**AC: We referred to the original Climada paper. We will correct this.**

- Line 270: In this context, "less important" is a better term than "subordinate".

AC: Ok

**Results**

Table 4: Wouldn't it make more sense to have the return period in the second column or integrated into the first column because it is how the scenarios were defined?
 AC: will be integrated in the first column

**AC: will be integrated in the first column**

- Line 319: This reference to Fig. 9a seems unnecessary. Potentially this is a mistake and should refer to Fig. 10a instead.

**AC: Ok will be corrected**

- Line 321: I think that adding "The NON-AGGREGATED average annual values ..." to this sentence would improve clarity.

**AC: Ok**

- Line 326: I do not completely understand the last sentence of this paragraph. Please clarify.

AC: Ok

- Line 329: A reference to Fig. 10b is missing.

**AC: Ok, will be corrected**

- Line 333: The value 3.5e-7 is a density and not a count. See other comment on changing the y-axis in Fig. 10 to counts or proportions.

**AC: will be corrected**

- Line 338: For consistency, I think it would be better not to change the units for annual impact. Hence, it should be CHF 0.73 million and not 734.06 kCHF.

AC: Ok

- Line 338: In scientific writing, the term "significant" should only be used in the context of statistical significance. Use the terms "considerable" or "substantial" instead.

**AC: Ok**

- Line 344: The last sentence of the paragraph is not necessary.

AC: Ok

- Line 349: The first sentence of this paragraph is not necessary because you explained this already. See earlier comment on including the description of the calculation of the combined impact in the methods section.

AC: Ok

- Line 357: I don't think this reference to Table 4 is necessary. The scenarios are well established by now.

AC: Ok

 Line 375: There is no need for this first sentence as this information is described in the method section already. \$

AC: Ok

**Discussion**

- Line 409: It does not seem necessary to describe the derivation of the fracture depth and avalanche scenarios again. This is described in the methods section already.

**AC: Ok**

- Line 420: It is best to avoid shortened forms in scientific writing (e.g., can't, isn't, etc.).

AC: Ok

- Line 436: I believe that "avalanche area" should be "avalanche release area" or even "potential release area". Please use consistent terminology throughout the manuscript.

**AC: Ok**

- Line 449: I believe it should say "structural weak points" or "structural weaknesses" but not "structural weakness points".

AC: Ok

- Line 470: I do not understand what you mean with "... out of their focus...".

AC: will be explained more precisely

**Discussion**

Your discussion section is currently almost exclusively a description of the potential limitations of your modelling approach. At the end, you provide a discussion of previous studies carried out in this field (L465-476), but it is rather brief and superficial. At the same time, some of the sections included in the result section seem to have more of a discussion character. Examples include the description of the spatial patterns that emerge from the analysis of the expected annual impact for individual objects (i.e., Fig. 9) and the discussion of the decreasing average annual impact with increasing return periods.

I think the discussion section could potentially be strengthen considerably by expanding it and reorganizing the material in the following fashion: 7

1) Move the discussion-like paragraphs from the results section into the discussion and combine them into a subsection that discusses the generalizable insight from the analysis beyond the study-site specific results.

2) Expand the comparison with existing research in this area to better highlight the strengths of your approach (some of this is currently included in the conclusion section) and how it expands on the previously existing methods.

3) Finish with a slightly tighter discussion of the limitations that highlight future research opportunities.

I think a structure like this would considerably strengthen the discussion section and the scientific contribution of your paper.

AC: We will reorganise this section according to all comments of both referees. We thank referee 1 very much for the detailed review, which will to improve our manuscript.

**Referee Comment 2 (RC2) by anonymous referee: November 10, 2022 Review:**

**1. Overview**

The paper by Ortner et al. describes a framework for spatially evaluating risk on a regional/country wide scale. The framework encompasses probability of release combined with hazard assessment, exposure and vulnerability using the risk assessment platform CLIMADA. The hazard part is evaluated using RAMMS:LSHIM with geometrically computed probable release areas for three different prototypical avalanches (1/30 yr, 1/100 yr and 1/300 yr return periods). This allows the authors to express the hazard in terms of approximated impact pressures. Further techniques are used to classify forested areas and adjust the simulationparameters accordingly. The exposure is evaluated using an identification process for building type in order to spatially represent monetary assets. Specific damage functions are then used to estimate the vulnerability of the buildings and the annual monetary impact estimated (using the software EconoMe) for each building. Alongside this, the aggregated annual monetary impact is calculated for the full evaluated area at a regional scale. Further to this, uncertainty and sensitivity analyses were conducted to assess the variability in input parameters.

The study presented explores an interesting topic with far reaching use cases. I believe the work to be of interest to NHESS readers. The study has a clear purpose and combines established techniques in recent literature for a novel framework for quantitative avalanche risk assessment. Due to this I believe the manuscript will be well suited for publication pending major revisions which the authors should address. These revisions fall into two major categories:

1) Technical detail:

There are a few areas of the text where further explanation is required. Specific examples are listed below in the detailed comments section. Due to the nature of the approach combining several established methodologies it is understandable that full explanation of each of the methods is not completely contained in the manuscript and is appropriately linked to citations, however, I feel in some cases it would be beneficial for the reader to get a bit more insight into the methods used so the paper can stand alone.

2) Structure and writing:

There are a lot of examples of typography errors and I feel there are a couple of sections where the structure could be adapted to have established the key concepts/methods prior to the section. The authors should revise the manuscript taking care to catch mistakes and organize the structure to better relay the scientific message of the paper.

Following up on these comments the authors should address, I have provided some more detailed explanation expanding on the examples where I have spotted some technical issues. I hope that the following comments are useful for the revision of your paper.

**2. Detailed comments**

**2.1 General**

• Throughout the manuscript there are a variety of typography errors that should be addressed. These fall into the following categories:

- Figures referenced out of order - the current order goes figure 1, 5, 3, 2 . . . . If appropriate figures should be placed in a strategic order such that the reader isn't required to jump around the paper too much as the contents are discussed in the text.

AC: We will make the adjustment so that all images appear in the text flow as they appear in the manuscript sequence.

- Typos.

AC: Will be corrected

- Some citations in text are wrong type - e.g. lacking parenthesis.

AC: Will be corrected

- Some more care needed with sentence structure wrong use of commas, missing words.
- AC: Will be corrected following the detailed instructions from point 2.2- 2.6 below
- Structure issues e.g. forest section refers to a lot of definitions that haven't been introduced yet and are not linked in the text e.g. could say "as expanded upon in section . . . " or adjust the structure of the paper such that the forest section comes after the required definitions for example the definition of three avalanche scenarios.

AC: Since both referees in RC1 and RC2 have comments on the Forest Layer Section, it will be revised and reorganised and all detailed comments of both referees will be addressed. Fig.2 will be removed according to RC1.

- Excessive abbreviations / acronyms not all are needed and inconsistent formatting and use within text for example some are not uniformly applied from figure title to the figure caption. Most are all uppercase, but some are lower case most are introduced more than once and then not used or the definition repeated or the same definition applied to two different variables (see e.g. Line 221). This is distracting to the reader rather than helpful! I recommend using a consistent heuristic for usage throughout and consider carefully in which cases encoding terms in an abbreviation is beneficial.
- AC: Agreed and will be corrected
- Units should not be in italics see e.g. m3 vs m3.

AC: OK

Some references in bibliography are missing information and are not of uniform style.

**AC: Will be corrected**

**2.2 Introduction**

- Line 24: Apparent missing word in sentence "In winter 2019, for example, exceptional snowfall events [occurred?] which caused high damages".

**AC: Will be corrected**

Equations (1) and (2) - although give a clearer explanation for the risk formulation used - are lacking in descriptive detail. For example fimp is undefined in the text at the equations and not further referenced. It may also be clearer to the reader if the syntax fimpact is used. Risk is repeatedly referred to as a product of hazard, exposure and vulnerability, however, equations suggest this is not the case - rather - it is a function of hazard, exposure vulnerability and probability of consequence. Probability of consequence is also undefined in the text and needs further explanation.

AC: Since the risk platform" Climada" was used, all equations and definitions of the term "risk" as well as the risk concept itself were strictly taken from-, and applied according to: Aznar-Siguan, G. and Bresch, D. N.: CLIMADA v1: a global weather and climate risk assessment platform, Geoscientific Model Development, 12, 3085–3097, https://doi.org/10.5194/gmd-12-3085-2019, 2019. The same syntax for fimp as in Aznar-Siguan and Bresch 2019 was used. If this should not be clear for the reader, it can be defined and worded as well as cited more precisely here.

 Line 41: introduces the idea of vulnerability under the IPCC definition with "economic, material or environmental consequences". The paper then utilizes this concept for vulnerability, however, appears to neglect the environmental consequence and focuses on economic and material consequences. While I think this choice does not affect the scientific contribution of this paper it could be expressed clearer if environmental consequences are intended to be included/neglected and why in the risk framework

AC: The core of this study is the presentation of the risk framework for buildings in order to provide a tool for decision makers in hazard prone regions for the field of natural hazards. The Climada platform would theoretically also allow the implication of environmental impacts. Since the avalanche process is a natural hazard process (compared to non-natural processes such as nuclear reactor accidents or man-made chemical hazards), the impact on human inhabited areas (buildings) seems to us to be of greater importance than the impact of avalanches on the environment and its ecosystems. Still, we can briefly add a paragraph to the study to explain why we have focused on economic and material impacts.

- Line 55: sentence unclear - extra comma?

**AC: Will be corrected**

- Figure 1: abbreviations in input key are undefined. Perhaps expand in the caption or link to text. AC: Will be expanded and corrected

**2.3 Methods**

- Line 84: Fig 5 referenced out of order.

AC: Will be corrected

Line 91: Sentence 1 of section 2.1.1 currently doesn't make sense - how does "surface roughness" influence the snowpack structure? I believe a different meaning was intended here. Further to this the concept of surface roughness should be expanded upon here - do you mean of the basal topography or of the snowpack itself? Also, which "transition" is unclear - do you mean the transition to decelerating material - or a specific transition in the topography?

AC: We will change Line 91 for better understanding to: "Forest influences the snowpack structure by interception and changed micro-climate as well as they increase the basal topographic surface roughness, and thus can prevent or reduce avalanche formation. Forest is able to stop movement of small avalanches due to higher friction in the avalanche path- and the avalanche deposition zone (Schneebeli and Bebi, 2004; Bebi et al., 2009; Teich et al., 2012; Brožová et al., 2021)" With avalanche transition zone we meant the avalanche path. Normally used in natural hazard community for hazard processes as "release zone (zone of origin) – transition zone (avalanche path) – deposition zone (runout zone)". See picture below by Bühler at al: https://nhess.copernicus.org/articles/20/1783/2020/.

- Figure 2: Typo for "treeheight". VHM introduced twice - once as treeheight and once as vegetation height -are these equivalent or does vegetation height also include shrubforest? This should be clarified. Avalanche disposition is also not well explained as to how it is included as a percentage value. This also needs to the be clarified.

AC: Typo will be corrected. Treeheight is originating from the so-called vegetation height model (short VHM) that's why the (VHM) is here, so its not introduced as vegetation height. Syntax and description of fields (treeheight/vegetation height) in Fig.2 are equally used as in the original paper of Bebi at al. 2021 but we understand the misleading and slightly confusing nomenclature. Also, according to referee 1, Fig.2 should be deleted. This might solve the issues and avoid further confusion.

- Line 114: Extreme and frequent scenarios not introduced yet - text should link forward to section 2.1.3 or consider restructuring.

**AC: will be corrected.**

- Line 114: Text should also be more clear on definitions and forest inclusion for all eight potential scenarios e.g. frequent / extreme - tiny small medium and large and should not leave it ambiguous to the reader.

AC: frequent and extreme are return period (rp) scenarios (30y rp = frequent, 100y and 300y rp are extreme), while tiny, small, medium and large are avalanche release sizes within a certain return period scenario (e.g. avalanche size tiny in the 300y rp scenario. All according to Bühler at al. 2018. We will correct that in the text for further clearance.

**- Line 141: Typo "statistic"**

AC: will be corrected

Line 153: Function for ψ stated is incorrect - see Salm, Burkard & Gubler (1990) - which states it as f(α) =0.219 sin(α) – 0.202 cos(α) (2.1) It should be checked that the correct form is applied in the ArcGIS code also. The function for the fracture depth d0 is then d0 = d\*0f(α), which should be further explained.

AC: We are sure the form of the formula we've applied in GIS  $\psi = 0.291/\sin(\alpha) - 0.202 \times \cos(\alpha)$  with  $\alpha$  being the slope angle is correct. 0.219 is a typo and will be corrected to 0.291. We will explain d0 = d\*0f( $\alpha$ ).Screenshot from Salm, Burkard & Gubler (1990) "Berechnung von Fliesslawinen Eine Anleitung für Praktiker mit Beispielen" page 6

Screenshot from «Teil III Anleitung zur Berechnung von Fliesslawienen» Urs Gruber, Perry Bartelt und Stefan Margreth 1999

Neigungsabhängigkeit  $f(\psi)$

$$f(\psi) = \frac{0.291}{\sin \psi - 0.202 \cdot \cos \psi}$$
 [] (III.4.3)

- Line 170: An overview of the how RAMMS:LSHIM differs from RAMMS:AVALANCHE should be included in this section.

AC: We can describe that there are no differences between the models but that RAMMS::LSHIM is simply a version that is able to process large amount of data on a very large scale.

Line 195: And overview of the EconoMe methodology should be explained.

AC: Complete overview of the used EconoMe methodology in Table 2. Will be corrected by referencing to Tab. 2 in Line 195

- Table 2: Typo in caption - swisstopo repeated.

AC: will be corrected.

Line 221: mean degree of damage = to mean percentage of damage?

AC: yes. We will remove the second misleading parenthesis and the MDD after "the mean percentage of damage (MDD)"

 Line 223: What is the conceptual difference between mean damage degree and mean damage ratio at a certain pressure?

AC: This specific expression is used in Climada and introduced in Aznar-Siguan and Bresch, 2019 and the CLIMADA online documentation: (MDD) gives the percentage of an exposed asset's numerical value that's affected as a function of intensity, such as the damage to a building by avalanches related to its value. The Percentage of Assets Affected (PAA) denotes the fraction of exposures that are affected, such as the number of buildings affected in an avalanche path (in this study its 100%). The Mean Damage Ratio (MDR) is the average impact to an asset. As 100% of buildings in an avalanche path are assumed to be affected MDD is equal to MDR. We will add this explanation for further clarity in the manuscript.

Figure 7: General question on the piecewise-step form of the impact functions - do we really expect a 29 kPa avalanche to give 40% impact but 31 kPa to give 80%? I guess my question lies in the idea that while the piecewise plot gives the context for the three prototypical avalanche scenarios wouldn't a smooth function be more realistic where the impact is interpolated between the three key thresholds?

AC: We are aware of the weaknesses of the stepwise impact functions and the uncertainties coming with it. These step functions come from the standardised classic hazard mapping in Switzerland where continual avalanche pressures are divided into three pressure classes (0-3kPa, 3-30kPa, and >30kPa) and thus are represented on standardised hazard maps. The use of these step functions is therefore a Swiss standard based on an evaluation of building damages and established by expert judgement. If continuous functions were applied now, there would still be no scientific basis for this and a study would first have to be carried out to describe and confirm the damage to different types of buildings with continuous functions. Such a study is definitely of interest to us but is beyond the scope of this paper. Therefore, we decided to use the Swiss standard of step functions following the EconoMe methodology, which is also currently in use by federal agencies and engineering offices all over Switzerland.

- Line 229: Citation missing parenthesis. Happens several times in this section and section 2.4.2.

**AC: will be corrected**

**2.4 Results**

- Line 313: Typo "at this locations".
- AC: will be corrected
- Table 4: Should explain how the combined scenario is calculated
- AC: will be corrected
- Figure 9: Black dot at Altdorf label leads to confusion is this a 107 CHF building? If so then aggregated value estimates don't make sense if not the symbol should be removed. In panel (a) the label also covers points, which should be corrected.
- AC: it's the centre dot (symbol) of Altdorf as it is a standard of open street map, will be removed
- As a general point about figure 9 the 1/30 yr expected annual impact seems pessimistic it would be
  interesting to see how this compares to actual spending in this area, say from insurance records / payouts or knowing which buildings have been damaged in the last 30 yrs 50 yrs.

AC: In this study we have used avalanche hazard indication mapping on large scale, this shows all possible avalanche hazards in the entire region and the associated risk, which are conservative by definition. Generally, this do not necessarily mean that all these avalanches will actually cause damages or have caused damages in the past. It is therefore a risk indication mapping on a very large scale to identify risk hotspots. A comparison to individual avalanches with insured values and actual losses is certainly interesting. However, there are many reasons why this isn't feasible at large scale: It would require that all buildings have the same monetary value as we have assumed in our large scale approach. The building layer is based on a methodology for creating an overview and not for specific building values (as this information isn't publicly available). Comparisons and evaluations could thus only be made on the basis of single avalanches and single objects and would thus miss the goal of large scale risk indication mapping, which is the main scope of our study.

- Figure 10: Typo "one eight-thousand randomly pulled samples" doesn't make sense. Also missing units for impact range.
- AC: will be corrected
- Line 363: missing CHF?
- AC: will be corrected

**2.5 Discussion**

- Line 453: "We consider buildings as point objects . . . " - how does this connect with the mapping - do you consider the building as the point at center of the building area/extent? Can this play a role with larger buildings say large barns or warehouses in the studied regions?

AC: To keep the model performant over a large areas, we use central points for each building in the centre of a buildings area (centroids). Large buildings are thus spatially underrepresented or equalised with small buildings: the exposed value, however, naturally increases with the size, e.g. with the volume or average number of people. We will add a paragraph to further explain this limitation.

- Line 468: "small scale of hazard process" - this is a bit ambiguous. Please clarify the scale intended.

AC: compared to windstorms or floods, avalanches are a rather small-scale process. We will explain it in more detail for better understanding.

- Line 473: "night light assessments" is not a common term, please clarify. AC: will be explained in more detail

**2.6 Conclusion**

- Line 482: "previously unknown threats"... - clarification needed. AC: We simulate all potential avalanches, according to the existing terrain characteristics. This results in modelled avalanches at places, where no damage has occurred so far. Therefore, so far unknown avalanches which potentially might cause damage, are modelled. We will formulate this in greater detail to create more clarity.

**2.7 Bibliography**

Salm, B, Burkard, André & Gubler, H U 1990 Berechnung von fliesslawinen. Eine Anleitung fuer praktiker mit beispielen. Tech. Rep.. Eidg.\ Institut\ für Schnee- und Lawinenforschung, CH–7260 Davos-Weissfluhjoch, publication Title: Mitteilungen des Eidgenössischen Instituts für Schnee und Lawinenforschung Volume: 47.4 AC: will be added to bibliography

The authors would like to thank the referee 2 for the detailed review and the useful comments. These will certainly help us to improve our manuscript!

---

## Referee Report (RR1)

**Large-scale risk assessment on snow avalanche hazard in alpine regions**

Ortner, G.[1,2,3], Bründl, M.[1,2], Kropf, C. M.[3,4], Röösli, T.[3,4] and Bresch, D. N.[3,4]

[1]WSL Institute for Snow and Avalanche Research SLF, 7260 Davos Dorf, Switzerland

[2]Climate Change, Extremes and Natural Hazards in Alpine Regions Research Center CERC, 7260 Davos Dorf, Switzerland

[3]Institute for Environmental Decisions, ETH Zurich, Universitätstr. 16, 8092 Zurich, Switzerland

[4]Federal Office of Meteorology and Climatology MeteoSwiss, Operation Center 1, P.O. Box 257, 8058 Zurich-Airport, Switzerland

Journal: *Natural Hazards and Earth System Sciences*
Manuscript id: https://doi.org/10.5194/nhess-2022-112

February 24, 2023

**Review:**

The authors have taken on board the comments from the previous round of revision, leading to a paper that I believe to be of wide interest to NHESS readers. The study has a clear purpose and combines established techniques in recent literature for a novel framework for quantitative avalanche risk assessment. Due to this, I believe the manuscript will be well suited for publication in NHESS.

I have included below some very minor typos that I consider to not effect the scientific content of the paper.

**1. Typos**

Paragraph indentation inconsistent through manuscript.

Figure 2 - Caption - capital O in Overview.

Line 169: (GEV - MLE) to (GEV-MLE)

Line 184: Remove . before $\psi$

Line 300: "This method allowed [us] to ..." → "This method allowed the creation of ..."

Line 442: Remove , after identifies

Line 443 and Line 449 : brackets around "mdd" → "(mdd)"

---

## Referee Report (RR2)

[revised manuscript text omitted]

265  [..[73] ]), which is [..[74] ]

[..[75] ][..[76] ][..[77] ][..[78] ][..[79] ][..[80] ][..[81] ][..[82] ][..[83] ][..[84] ][..[85] ][..[86] ][..[87] ]

[..[88] ][..[89] ][..[90] ] [..[91] ][..[92] ][..[93] ][..[94] ][..[95] ][..[96] ][..[97] ][..[98] ][..[99] ][..[100] ][..[101] ][..[102] ][..[103] ][..[104] ][..[105] ][..[106] ]
* * *
[73]removed: Snow depth increase data was taken from the SLF-station Meien ME2 (see Fig. 4). This station has a time series of 66 years and includes extreme snowfalls (e.g. winter 1950/51 and 1998/99) making the data basis more reliable for extreme events. It is located in the center of the project area at an altitude of 1320 m a.s.l.

[74]removed: close to the average altitude of the release areas. Since the snow depth increase $\Delta HS(3)$ (see Fig. 3) at the meteorological station Meien is taken in flat field, a standard slope inclination correction (Margreth, 2007) was conducted at all release polygons to consider an adapted fracture depth. In order to correct for the influence of drifting snow, a snow drift factor was added depending on the size of the scenario (Salm et al., 1990). In practice, this factor strongly depends on local conditions at the release zones (Margreth, 2007). For the 30-year return period scenario we added 30 cm and for the 100y return period scenario 40 cm and the 300y return period scenario 50 cm of drifting snow correction. After defining the fracture depth and the area of the release zones as well as applying the corresponding correction factors, the volume for each release area could be determined and used for the hazard simulation with the RAMMSlarge scale hazard indication mapping model (Bühler et al., 2018, 2022).

[revised manuscript text omitted]

**4.3   General insights and strengths of the approach**

605    In this work, we have linked a risk tool with a new method for large scale hazard assessment. We have calculated and presented avalanche risks of over 40,000 single avalanches in different hazard scenarios for more than 13,300 buildings over an area of 469.3 km$^2$. Due to the complexity and the small scale of the hazard process (compared to storms or floods), avalanche risk assessment is often addressed at the local level. Studies previously carried out in this field, such as [..366 ]those of Fuchs et al. (2004) and Fuchs et al. (2006) or more recent studies such as Zgheib et al. (2020) have put their focus on the scale of

610    particular villages [..367 ]and are depending on existing hazard maps. Ettlin et al. (2014) established a risk tool for buildings using RAMMS::Avalanche on single slopes. Large scale approaches as the one of Kazakova et al. (2017) assess risk for 60
* * *
358 removed: weakness points
359 removed: development of these steps
360 removed: real damage always follows
361 removed: this depends
362 removed: This allows
363 removed: the
364 removed: at the exact
365 removed: weaken
366 removed: that
367 removed: .

buildings, others [..[368] ]focus on risks and [..[369] ]protection forest development such as Renner and Steger (2021). Fuchs et al. (2015) uses detailed avalanche hazard maps at regional scale and flood hazards [..[370] ]at a countrywide scale. [..[371] ]Bühler et al. (2019) focus on large scale avalanche mapping from satellite data after high intensity snowfall events, but do not address corresponding risk assessment for objects in an entire region. Several studies and expert appraisals were conducted by other authors in the Gotthard area where our case study is located. Most of them are detailed expert judgments on single slopes to assess avalanche risk for specific avalanche paths or specific objects at risk. Margreth and Ammann (2004) conducted a study on the south side of Lukmanier Pass in the central part of the Swiss Alps to develop hazard scenarios to describe the impact of avalanches on specific structures such as bridges. Margreth et al. (2003), for instance, conducted an avalanche risk analysis using previous versions of the RAMMS avalanche simulation software. They also conducted a case study in the Gotthard region with the focus on avalanche safety on pass roads, but based on a detailed single slope hazard evaluation (Margreth et al., 2003). While such approaches are of high detail and accuracy, they do not allow for large-scale or site-independent application. In contrast, our method allows for modelling avalanche danger for unlimited sized territories with little time expenditure, without using hazard maps previously provided by local authorities. Exposure values can [..[372] ]either be generated with existing data or roughly estimated in CLIMADA via night light assessments if no detailed information on buildings is available (Aznar-Siguan and Bresch, 2019). Nightlight assessment takes the light intensity of satellite images taken of the landscape at night and assigns monetary values to a location depending on the light intensity. This approach allows the method to be applied worldwide even with a limited exposure database available. The RAMMS::LSHIM method and the good adaptability of the risk tool is a significant advantage when applying this method in areas [..[373] ]with no or limited hazard or asset information available.

**5 [..[374] ]**

Our novel approach for simulating avalanches and assessing the spatially distributed risk through monetary valuation is valid at large scales with a high degree of detail (single-object-resolution). Owing to the avalanche simulations from the RAMMS:LSHIM method, no external avalanche hazard information is needed. [..[375] ]CLIMADA is easy to adapt and already being used for a variety of other hazards such as hurricanes and winter storms, and now avalanches. This concept, in combination with large scale hazard mapping, can easily be [..[376] ]used for different hazards [..[377] ]such as rockfall, debris flows or landslides. Additionally, other exposure scenarios such as traffic routes or  or critical
* * *
[368]removed: out their

[369]removed: forest (Renner and Steger, 2021)

[370]removed: on

[371]removed: If one compares the method we use, avalanche danger can be modeled

[372]removed: also be generated

[373]removed: where there is

[374]removed: Conclusions and outlook

[375]removed: Additionally, this concept

[376]removed: adapted

[377]removed: or exposure scenarios

[revised manuscript text omitted]